# LET THE CODE LLM EDIT ITSELF WHEN YOU EDIT THE CODE

**Zhenyu He** [*♡]   **Jun Zhang** [♠]   **Shengjie Luo** [♡]   **Jingjing Xu** [♠]   **Zhi Zhang** [♠]   **Di He** [♡†]

[♡]National Key Laboratory of General Artificial Intelligence,
School of Intelligence Science and Technology, Peking University
[♠] ByteDance Inc.

## ABSTRACT

In this work, we investigate a typical scenario in code generation where a developer edits existing code in real time and requests a code assistant, e.g., a large language model, to re-predict the next token or next line on the fly. Naively, the LLM needs to re-encode the entire KV cache to provide an accurate prediction. However, this process is computationally expensive, especially when the sequence length is long. Simply encoding the edited subsequence and integrating it to the original KV cache meets the temporal confusion problem, leading to significantly worse performance. We address this efficiency and accuracy trade-off by introducing **P**ositional **I**ntegrity **E**ncoding (PIE). Building upon the rotary positional encoding, PIE first removes the rotary matrices in the Key cache that introduce temporal confusion and then reapplies the correct rotary matrices. This process ensures that positional relationships between tokens are correct and requires only a single round of matrix multiplication. We validate the effectiveness of PIE through extensive experiments on the RepoBench-C-8k dataset, utilizing DeepSeek-Coder models with 1.3B, 6.7B, and 33B parameters. Our evaluation includes three real-world coding tasks: code insertion, code deletion, and multi-place code editing. Results demonstrate that PIE reduces computational overhead by over 85% compared to the standard full recomputation approach across all model sizes and tasks while well approximating the model performance. Code is available at `https://github.com/zhenyuhe00/PIE`.

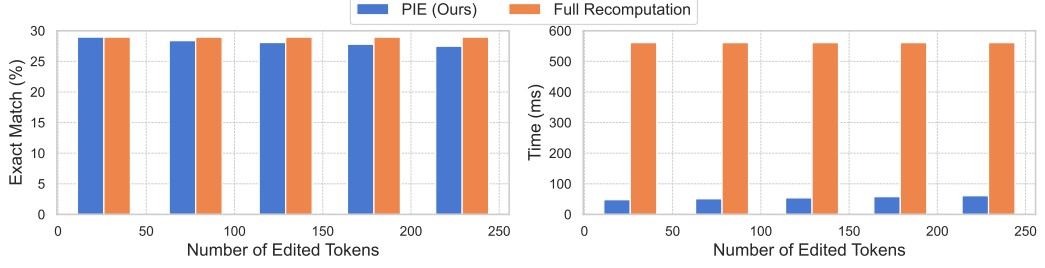

Figure 1: Latency and accuracy comparison of the full recomputation approach and our PIE using DeepSeek-Coder 6.7B on the RepoBench-C-8k(XF-F) Python dataset on a single A100 GPU. The latency only records the time cost for the KV cache update.

## 1 INTRODUCTION

Large language models (LLMs) (Dettmers et al., 2022; Anil et al., 2023; Touvron et al., 2023; Zeng et al., 2023) have seen widespread adoption and achieved impressive results across various natural language processing (NLP) tasks. Despite these successes, LLMs face significant computational

---

[*]Work done during Zhenyu's internship at ByteDance.
[†]Correspondence to: Di He<dihe@pku.edu.cn>

challenges, particularly in handling long sequences. To address this, numerous approaches have been proposed to accelerate the inference process, including lossless (e.g., memory and IO optimization (Dao et al., 2022; Kwon et al., 2023; Sheng et al., 2023), speculative decoding (Stern et al., 2018; Leviathan et al., 2023)) and lossy techniques (e.g., quantization (Frantar et al., 2022; Xiao et al., 2023) and KV cache eviction (Xiao et al., 2024; Zhang et al., 2024)). We refer to the above setting as the *static* setting, where the content is fixed, and the goal is to generate responses efficiently without compromising too much on performance.

Besides the static setting, we observe there is a strong demand for an alternative, which we call the *real-time editing* setting, where users frequently edit the content and expect the LLM to generate correct responses based on the updated information. A typical scenario is the interactive coding assistant, where developers often make incremental changes to their existing code and require the AI copilot to correctly predict the next line or complete a partial code snippet on the fly. The standard approach is re-encoding the KV cache of the content after each edit and then making the prediction. However, as illustrated in Figure 1, this approach leads to considerable substantial computational overhead and latency when the content is long (Fu, 2024; Agrawal et al., 2024), making it impractical for real-time applications where quick and accurate responses are essential.

In this paper, we aim to improve the efficiency of AI copilots in real-time editing scenarios, as illustrated in Figure 2. Naively, an efficient strategy is encoding only the edited subsequence and then directly integrating those keys and values into the original KV cache. However, this strategy results in temporal confusion between the pre-edit and post-edit sequences. Keys of certain positions either disappear or multiple keys at different positions share the same index, causing the model to attend to incorrect information, leading to poor next-token prediction performance in practice. To address this problem, we introduce **P**ositional **I**ntegrity **E**ncoding (PIE). PIE is built upon rotary positional encoding (RoPE) (Su et al., 2021), the de-facto standard component in modern LLMs. PIE first removes the rotary matrices in the Key cache that introduce temporal confusion and then reapplies the correct rotary matrix for each position through simple matrix multiplications. By ensuring that the positional relationships between tokens in the new sequence are unique and consecutive, PIE can help the model make accurate predictions. It is worth noting that the calculation of PIE requires only a single round of matrix operations to modify the KV cache, resulting in negligible computational overhead.

We demonstrate the effectiveness of Positional Integrity Encoding (PIE) through extensive experiments conducted on the RepoBench-C-8k dataset, utilizing the DeepSeek-Coder (Guo et al., 2024) models with 1.3B, 6.7B, and 33B parameters. To rigorously evaluate PIE's performance, we curated three tasks designed to simulate real-world coding scenarios: code insertion, code deletion, and multi-place code edition. These tasks were chosen to reflect common operations that developers perform during interactive coding sessions, thereby providing a comprehensive assessment of PIE's practical utility. Our experimental results indicate that PIE achieves a reduction in computational overhead of over 85% for editing the KV cache across all model sizes and tasks without compromising performance compared to the naive full-recomputation approach. By leveraging PIE, developers can experience efficient interactions with AI coding assistants.

## 2 RELATED WORK

**Positional Encodings** Positional information is essential for modeling languages. The original Transformer model (Vaswani et al., 2017) encodes positional information using Absolute Positional Encoding (APE). In particular, a (learnable) real-valued embedding is assigned to each position $i$. Differently, Relative Positional Encodings (RPE) (Shaw et al., 2018; Dai et al., 2019; Raffel et al., 2020; Press et al., 2022; Su et al., 2021; Luo et al., 2021; 2022; Chi et al., 2022; Sun et al., 2023; Chi et al., 2023; Li et al., 2023; He et al., 2024) instead encode the relative distance $i - j$ for each position pair $(i, j)$. One of the most widely used RPE in state-of-the-art LLMs is Rotary Position Encoding (RoPE) (Su et al., 2021). RoPE rotates the query and key vectors by an angle proportional to their absolute positions before the attention mechanism, resulting in the attention being a function of the relative distance between tokens.

In the literature, relative positional encodings play essential roles across various tasks and data modalities, such as improving the length extrapolation capability of language models (Press et al., 2022; Sun et al., 2023; Chi et al., 2022; 2023; He et al., 2024) and enabling flexible modeling of

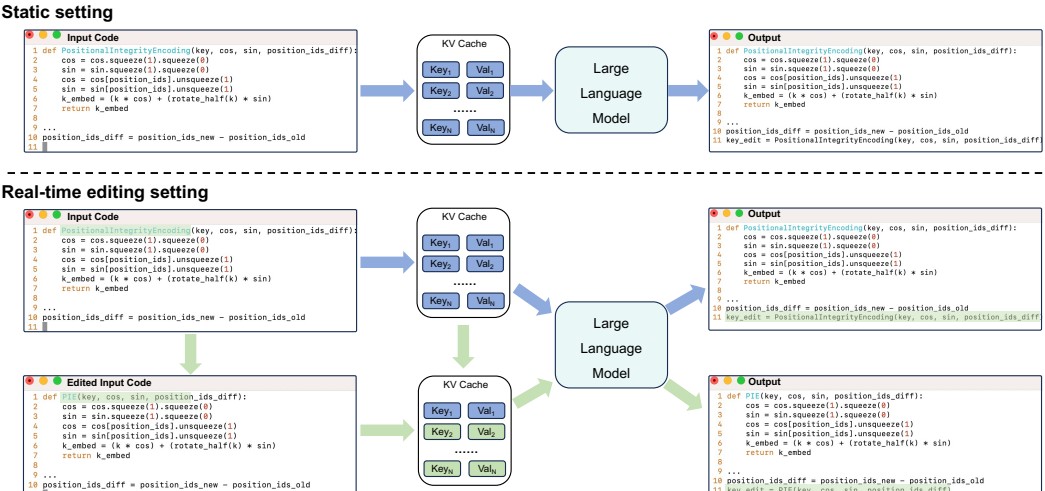

Figure 2: Illustration of the KV cache mechanism in both static and real-time editing settings for large language models (LLMs). **Top:** In the static setting, the model processes a fixed input to generate predictions, leveraging precomputed Key/Value (KV) pairs stored in the cache. **Bottom:** In the real-time editing setting, the input code is frequently edited, necessitating updates to the KV cache to maintain accurate information to generate the correct next tokens. Our objective is to optimize the efficiency of the green arrow pathway, which represents the process of updating the KV cache in response to code edits.

structural information beyond sequence data like images (Liu et al., 2021) and graphs (Ying et al., 2021; Zhang et al., 2023a; Luo et al., 2023). In this work, we develop the Positional Integrity Encoding based on RoPE to improve the efficiency of LLMs in the real-time editing setting.

**Transformer Efficiency** Improving the efficiency of Transformer models has great significance in real-world applications. In the literature, existing approaches can be briefly categorized into (1) efficient attention, (2) model compression, and (3) system-architecture co-design.

The attention module of Transformer needs to calculate pairwise correlations between all positions, resulting in quadratic time and memory cost with respect to the sequence length. To reduce the cost, many efficient attention variants have been proposed, such as (1) sparse attentions (Child et al., 2019; Beltagy et al., 2020; Qiu et al., 2020), which design either pre-defined or learnable patterns to reduce the amount of the key-value pairs that each query needs to attend to; (2) approximation-based attention (Katharopoulos et al., 2020; Wang et al., 2020; Choromanski et al., 2021; Kitaev et al., 2020; Tay et al., 2020; Roy et al., 2021), which use tailored approaches like low-rank projection or random features to approximate standard attention for efficient computation.

Another perspective for Transformer efficiency is model compression, mainly including (1) pruning (Wang et al., 2021; Hubara et al., 2021; Ma et al., 2023; Frantar & Alistarh, 2023), which aims at removing redundant model parameters or layers for efficient deployment without scarifying performance; (2) quantization (Yao et al., 2022; Park et al., 2022; Dettmers et al., 2022; Frantar et al., 2022; Xiao et al., 2023; Liu et al., 2023), which uses post-processing to represent weights and activations via low-precision format for reducing time and memory costs; (3) knowledge distillation (Sanh et al., 2019; Gu et al., 2024), which uses a smaller model to learn knowledge from a large model for balancing efficiency-accuracy trade-offs.

In the era of LLMs, the importance of system-architecture co-design has been highlighted to improve Transformer efficiency further. Many works begin to design more efficient approaches for serving Transformers regarding the characteristics of computer systems for real-world applications, such as FlashAttention (Dao et al., 2022; Dao, 2023), PagedAttention (Kwon et al., 2023), and FlexGen (Sheng et al., 2023) that are proposed for memory and I/O optimization. Additionally, to reduce functional calls during generation, speculative decoding (Stern et al., 2018; Leviathan et al., 2023; Chen et al., 2023; Miao et al., 2023; Spector & Re, 2023; Cai et al., 2024; Zhang et al., 2023b;

He et al., 2023; Li et al., 2024) has been proposed. Our Positional Integrity Encoding is specially designed to improve the efficiency of LLMs in real-time editing settings, which can be seamlessly combined with all the above-introduced approaches to achieve further speed-ups.

## 3 METHODS

### 3.1 BACKGROUND

Let $s = (w_1, w_2, \ldots, w_n)$ represent the input token sequence, where each $w_i$ belongs to a fixed vocabulary. Let $\theta_{\mathrm{LLM}}$ represent a Transformer-based large language model, which can calculate the conditional probability distribution of the next token $p(w_{n+1}|s; \theta_{\mathrm{LLM}})$ and generate tokens iteratively. Typically, the input $s$ is fixed. Therefore, the generation process of LLMs usually employs the KV Cache mechanism (Pope et al., 2023) to store previously computed Key/Value vectors during each layer's attention calculation. We denote the KV cache as $\boldsymbol{K} = (K_1, K_2, \ldots, K_n)$ and $\boldsymbol{V} = (V_1, V_2, \ldots, V_n)$, where $K_i$ and $V_i$ are the keys and values associated with token $w_i$. When predicting the token at position $n+1$, we can use $w_n$ as input and, in each layer, compute the attention between the current hidden representation and the stored KV cache, avoiding recomputing the hidden representation of previous tokens. Without any confusion, we also denote the next token probability distribution as $p(w_{n+1}|\boldsymbol{K}, \boldsymbol{V}, w_n; \theta_{\mathrm{LLM}})$.

In this study, we aim to investigate a new scenario where the context $s$ is real-time edited by users, which makes it impossible for the KV cache to predict the correct next token without any modification. We can model such real-time context as a sequence of steps. Each step can be formulated as an action where tokens from position $i$ to $j$ in $s$ are edited, resulting in a modified sequence $todo$ represent the new inputs that replace $[w_i, \ldots, w_j]$. Our goal is to accurately and efficiently predict $w_{n+1}$ given by $\theta_{\mathrm{LLM}}$ on $s^{\mathrm{edit}}$. This problem is crucial in various scenarios. For instance, users can frequently edit their previous codes for different purposes and expect the code language model to swiftly adapt to these changes and predict the correct next line based on the updated information.

As the context between position $i$ and position $j$ is edited, the KV cache corresponding to these tokens must be updated. Furthermore, these changes will impact the representations of subsequent tokens after position $j$, thereby necessitating updates to all subsequent KV cache. The naive approach involves a full-recomputation strategy: re-encoding all the KV cache for $[a_1, a_2, \ldots, a_m, w_{j+1}, \ldots, w_n]$ layer by layer, followed by making predictions using the updated cache $\boldsymbol{K}^*$ and $\boldsymbol{V}^*$. This approach ensures the KV cache is exact when predicting the next tokens. However, it is easy to see that it is computationally expensive, especially when the edits are light but the texts to be re-encoded are long. It's worth noting that the original $\boldsymbol{K}$ and $\boldsymbol{V}$ already encode rich information on $[w_{j+1}, \ldots, w_n]$, and a full recomputation may not be essential for practical problems. With this in mind, we seek to find ways to efficiently edit $\boldsymbol{K}$ and $\boldsymbol{V}$, yielding $\boldsymbol{K}^{\mathrm{edit}}$ and $\boldsymbol{V}^{\mathrm{edit}}$, which approximates $p(w_{n+1}|\boldsymbol{K}^*, \boldsymbol{V}^*, w_n; \theta_{\mathrm{LLM}}) \approx p(w_{n+1}|\boldsymbol{K}^{\mathrm{edit}}, \boldsymbol{V}^{\mathrm{edit}}, w_n; \theta_{\mathrm{LLM}})$ in an effective way.

### 3.2 POSITIONAL INTEGRITY ENCODING (PIE)

When a user modifies $s$ into $s^{\mathrm{edit}}$, KV cache associated with the first $i$ tokens, i.e., $\boldsymbol{K}_{[1:i]}$ and $\boldsymbol{V}_{[1:i]}$, remains unchanged. As $[a_1, a_2, \ldots, a_m]$ is the user's new input, we feed this subsequence to the LLM to obtain the keys and values from position $i+1$ to $i+m$. We denote this piece of new KV cache as $\boldsymbol{K}^{\mathrm{edit}}_{[i+1:i+m]}$ and $\boldsymbol{V}^{\mathrm{edit}}_{[i+1:i+m]}$, and now have the edited KV cache as:

$$\boldsymbol{K}^{\mathrm{edit}} = \mathrm{Concat}(\boldsymbol{K}_{[1:i]}, \boldsymbol{K}^{\mathrm{edit}}_{[i+1:i+m]}, \boldsymbol{K}_{[j+1:n]}), \tag{1}$$

$$\boldsymbol{V}^{\mathrm{edit}} = \mathrm{Concat}(\boldsymbol{V}_{[1:i]}, \boldsymbol{V}^{\mathrm{edit}}_{[i+1:i+m]}, \boldsymbol{V}_{[j+1:n]}), \tag{2}$$

where the red symbols indicate real-time calculations.

**Challenges.** The key challenge lies in how to edit the succeeding KV cache $\boldsymbol{K}_{[j+1:n]}$ and $\boldsymbol{V}_{[j+1:n]}$. Clearly, the modification of $[a_1, a_2, \ldots, a_m]$ impacts the subsequent content in two ways: semantically and structurally. The semantic impact refers to the changes in the understanding of the subsequent text caused by the edited content. This can be a problem in natural language applications, such as dialog systems, where modifications to earlier conversations can significantly influence the

generation of current responses. The other impact is structural, primarily concerning the temporal confusion between the pre-edit and post-edit sequences when $j - i \neq m$. This issue arises with common editing actions in code, such as additions and deletions (corresponding to $j - i = 0$ or $m = 0$). To be more concrete, imagine the original sequence has 5 tokens. If we add three tokens between the second and third position, it will occupy the positional index $[3, 4, 5]$. We calculate $\boldsymbol{K}_{[3:5]}^{\text{edit}}$ and $\boldsymbol{V}_{[3:5]}^{\text{edit}}$ in equation (1) and (2). However, in the original $\boldsymbol{K}_{[3:5]}$ and $\boldsymbol{V}_{[3:5]}$, the positional index $[3, 4, 5]$ is also occupied. If we integrate them together and take no actions during the next token prediction, the model will calculate similarity with multiple keys in the KV cache with the same index $[3, 4, 5]$, causing confusion and potential prediction errors. Empirically, in code tasks, we find that the semantic impact is relatively small. Addressing temporal confusion for light edits alone can already lead to good performance (see the experiments in Section 4 for more details).

**Our approach.** To mitigate the temporal confusion during real-time editing, we propose a simple yet effective solution: **P**ositional **I**ntegrity **E**ncoding (PIE), which ensures that positional information remains correctly ordered after editing without the need to re-encode the KV cache for subsequent tokens. PIE builds upon the rotary positional encoding (RoPE) (Su et al., 2021), which is the most widely used positional encoding in LLMs. Without loss of generality, given a query vector $\boldsymbol{x}_i$ at position $i$ and a key vector $\boldsymbol{x}_j$ at position $j$, RoPE calculates the dot-product similarity using

$$z_{ij} = \boldsymbol{x}_i^T W_q^T \boldsymbol{R}_{j-i} W_k \boldsymbol{x}_j \tag{3}$$

where $\boldsymbol{R}_{j-i}$ is the rotary matrix parameterized by the relative distance $j - i$, and $W_q$ and $W_k$ are learnable projection matrices. By definition, $\boldsymbol{R}_{j-i}$ can be expressed by the multiplication of two rotary matrices:

$$\boldsymbol{R}_{j-i} = \boldsymbol{R}_i^T \boldsymbol{R}_j \tag{4}$$

For practical implementation, during inference, we compute $\boldsymbol{R}_{\Theta,i} W_k x_i$ as the key on the fly and store it in the cache, and when a query arrives at a new position, we rotate the query using its corresponding rotary matrix and calculate its similarity with all the keys in the cache to obtain the attention scores.

It can be easily seen that the positional information in the KV cache is encoded within the rotary matrix. When an edit occurs, the rotary matrix associated with the keys must be adjusted to reflect their post-edit locations. Leveraging the formulation of RoPE-based attention calculation, this challenge can be addressed by first removing the rotary matrices in $\boldsymbol{K}$ that introduce temporal confusion and then reapplying the correct rotary matrix. In detail, assume we would like to update the key vector $\boldsymbol{k}_{j'}^l$ for the original position $j' \in [j + 1, n]$, where $l \in [1, L]$ is the layer index. We can simply edit the key vector by using

$$\boldsymbol{k}_{j'}^{\text{edit},l} = \boldsymbol{R}_{i+m+j'-j} \boldsymbol{R}_{j'}^{-1} \boldsymbol{k}_{j'}^l \tag{5}$$

where $\boldsymbol{R}_{j'}^{-1}$, the inverse rotary matrix at position $j'$, is used to remove the incorrect positional information, and $\boldsymbol{R}_{i+m+j'-j}$ is used to encode the correct position $i + m + j' - j$ in $s^{\text{edit}}$. It can easily seen that the computation can be further simplified as

$$\boldsymbol{k}_{j'}^{\text{edit},l} = \boldsymbol{R}_{i+m+j'-j} \boldsymbol{R}_{j'}^{-1} \boldsymbol{k}_{j'}^l = \boldsymbol{R}_{i+m+j'-j} \boldsymbol{R}_{-j'} \boldsymbol{k}_{j'}^l = \boldsymbol{R}_{i+m-j} \boldsymbol{k}_{j'}^l \tag{6}$$

Hence, the full editing process for $\boldsymbol{K}_{[j+1:n]}$ is as follows:

$$\boldsymbol{K}_{[j+1:n]}^{\text{edit}} = [K_{j+1}^{\text{edit}}, \ldots, K_n^{\text{edit}}] \tag{7}$$

$$\text{where each } K_{j'}^{\text{edit}} = \{\boldsymbol{k}_{j'}^{\text{edit},1}, \cdots, \boldsymbol{k}_{j'}^{\text{edit},l}, \cdots, \boldsymbol{k}_{j'}^{\text{edit},L}\}, j' \in [j+1, n], l \in [1, L]$$

$$\text{each } \boldsymbol{k}_{j'}^{\text{edit},l} = \boldsymbol{R}_{i+m-j} \boldsymbol{k}_{j'}^l$$

Unlike the full recomputation approach, the above calculation only requires a single round of matrix multiplication to directly modify the pre-computed KV cache, where the computational overhead can be considered negligible. By utilizing these transformations, we finally construct the edited KV cache as:

$$\boldsymbol{K}^{\text{edit}} = \text{Concat}(\boldsymbol{K}_{[1:i]}, \boldsymbol{K}_{[i+1:i+m]}^{\text{edit}}, \boldsymbol{K}_{[j+1:n]}^{\text{edit}}), \tag{8}$$

$$\boldsymbol{V}^{\text{edit}} = \text{Concat}(\boldsymbol{V}_{[1:i]}, \boldsymbol{V}_{[i+1:i+m]}^{\text{edit}}, \boldsymbol{V}_{[j+1:n]}), \tag{9}$$

Table 1: Statistics of RepoBench-C-8k (Liu et al., 2024a) test set.

| Language | XF-F | XF-R | IF | Average Number of Tokens |
|---|---|---|---|---|
| Python | 18,000 | 7,500 | 10,500 | 3,967 |
| Java | 18,000 | 7,500 | 10,500 | 4,179 |

where the red symbols indicate real-time calculations. The LLM then makes predictions based on $p(x_{n+1}|\boldsymbol{K}^{\mathrm{edit}}, \boldsymbol{V}^{\mathrm{edit}}, x_n; \theta_{\mathrm{LLM}})$. It is worth noting that PIE is compatible with KV cache eviction methods (Xiao et al., 2024; Zhang et al., 2024; Liu et al., 2024b). These KV cache eviction methods focus on reducing the memory usage of the KV cache during inference. PIE is designed to obtain the KV cache of the edited context with minimal overhead. By integrating PIE with KV cache eviction methods, it is possible to maintain efficient memory management while ensuring the integrity of the positional information in the real-time edit setting.

## 4    EXPERIMENTS

In this section, we empirically study the effectiveness of our proposed method. In particular, we aim at answering the following questions through experiments:

- **Question 1**: Can our Positional Integrity Encoding maintain the prediction accuracy of full re-computation in code editing scenarios?
- **Question 2**: How much efficiency improvement can be achieved by using our Positional Integrity Encoding compared to existing approaches?
- **Question 3**: How large is the gap between our Positional Integrity Encoding and full re-computation in terms of LLM's predictions & representations?

We will answer each question with carefully designed experiments in the following sub-sections. We also conduct additional experiments in Appendix B.

### 4.1    EXPERIMENTAL SETUP

**Tasks.**    Our experiments are conducted on RepoBench-C-8k (Liu et al., 2024a). This benchmark focuses on the prediction of the next line of code, given a set of in-file context (including import statements and preceding lines before the target line), and cross-file context (comprising snippets from other files parsed by import statements). The detailed statistics of RepoBench-C-8k is shown in Table 1. To effectively evaluate next-line prediction performance of code LLMs, we follow Liu et al. (2024a) to use three task settings: (1) Cross-File-First (XF-F): mask the first appearance of a cross-file line within a file; (2) Cross-File-Random (XF-R): mask a random and non-first occurrence of a cross-file line; (3) In-File (IF): mask an in-file line that does not involve any cross-file modules. Moreover, we carefully design three real-world scenarios covering code insertion, code deletion, and code edition to comprehensively examine our approach. See Appendix A.1 for more detailed descriptions of tasks construction.

**Settings.**    In our experiments, we employ DeepSeek-Coder (Guo et al., 2024), a code LLM that achieves strong performance in handling repository-level code completion tasks (We also conduct experiments on CodeLlama (Roziere et al., 2023) in Appendix A.2). We use Transformers (Wolf et al., 2020) as our codebase. We benchmark our method on models of different sizes covering 1.3B, 6.7B, and 33B. During inference, the greedy decoding strategy is used to deterministically generate 64 tokens. For 1.3B and 6.7B models, all the experiments are conducted on a single NVIDIA A100 GPU. For 33B models, the time for encoding the context is conducted on two NVIDIA A100 GPUs and the full generation process is conducted on eight NVIDIA A100 GPUs. The first non-comment line in the output is truncated and used as the prediction. The batch size is set to 1. All experiments are repeated three times with different seeds and the averaged scores are reported.

**Evaluation.**    For comparison with our Positional Integrity Encoding, we choose two standard approaches as baselines: (1) Full-recomputation: re-compute the KV cache for all edited tokens

Table 2: **Performance comparisons of insertion experiments.** In this task, for each next-line prediction target, we insert several lines of code into its context randomly to simulate real-world scenarios. EM and ES denote the Exact Match and Edit Similarity score respectively. All results demonstrate that our Positional Integrity Encoding approach brings substantial speed-ups without performance drops.

| | Model | Method | XF-F | | | XF-R | | | IF | | |
|---|---|---|---|---|---|---|---|---|---|---|---|
| | | | EM | ES | Time | EM | ES | Time | EM | ES | Time |
| Python | 1.3B | Full-recomputation | 22.42 | 65.26 | 192ms | 35.41 | 72.96 | 193ms | 28.78 | 69.22 | 193ms |
| | 1.3B | Conflict Fast Encoding | 7.32 | 43.73 | 23ms | 10.61 | 47.18 | 22ms | 8.91 | 45.25 | 22ms |
| | 1.3B | PIE | 22.3 | 65.2 | 29ms | 35.33 | 72.88 | 28ms | 28.69 | 69.13 | 29ms |
| Python | 6.7B | Full-recomputation | 28.95 | 70.11 | 561ms | 40.89 | 76.19 | 564ms | 35.26 | 72.73 | 562ms |
| | 6.7B | Conflict Fast Encoding | 5.35 | 33.32 | 34ms | 6.52 | 35.25 | 34ms | 6.09 | 38.76 | 34ms |
| | 6.7B | PIE | 28.83 | 70.01 | 50ms | 40.77 | 76.14 | 50ms | 35.2 | 72.72 | 50ms |
| Python | 33B | Full-recomputation | 35.75 | 73.46 | 2194ms | 46.0 | 78.9 | 2199ms | 39.75 | 75.12 | 2194ms |
| | 33B | Conflict Fast Encoding | 3.96 | 30.13 | 126ms | 5.41 | 32.32 | 121ms | 3.92 | 35.56 | 127ms |
| | 33B | PIE | 35.77 | 73.45 | 134ms | 45.74 | 78.85 | 140ms | 39.74 | 75.1 | 141ms |
| Java | 1.3B | Full-recomputation | 26.21 | 70.89 | 200ms | 36.77 | 76.31 | 200ms | 45.89 | 78.04 | 198ms |
| | 1.3B | Conflict Fast Encoding | 0.29 | 3.12 | 22ms | 0.57 | 3.19 | 23ms | 0.7 | 2.63 | 23ms |
| | 1.3B | PIE | 26.13 | 70.82 | 30ms | 36.67 | 76.25 | 30ms | 45.92 | 77.99 | 29ms |
| Java | 6.7B | Full-recomputation | 32.51 | 75.56 | 578ms | 41.97 | 79.41 | 578ms | 50.86 | 80.53 | 578ms |
| | 6.7B | Conflict Fast Encoding | 0.47 | 2.77 | 34ms | 0.76 | 2.78 | 35ms | 0.67 | 2.48 | 33ms |
| | 6.7B | PIE | 32.21 | 75.47 | 50ms | 41.96 | 79.32 | 49ms | 50.85 | 80.43 | 48ms |
| Java | 33B | Full-recomputation | 35.05 | 76.93 | 2269ms | 44.95 | 80.87 | 2281ms | 53.23 | 81.76 | 2270ms |
| | 33B | Conflict Fast Encoding | 0.38 | 2.51 | 120ms | 0.59 | 2.40 | 122ms | 0.68 | 2.09 | 122ms |
| | 33B | PIE | 34.78 | 76.78 | 138ms | 45.01 | 80.95 | 133ms | 53.16 | 81.66 | 139ms |

and subsequent tokens; (2) Conflict Fast Encoding: re-compute the KV cache for the edited tokens while keeping the rest of the cache intact (i.e., using equation (1,2)). Following Lu et al. (2021), we use Exact Match (EM) and Edit Similarity (ES) (Svyatkovskiy et al., 2020) to evaluate the accuracy of the predicted code lines on code completion tasks. We also report the time required to encode the edited context for efficiency evaluation.

## 4.2 MAIN RESULTS

**Positional Integrity Encoding perfectly preserves the full re-computation performance.** Results of different code editing settings are presented in Table 2, 3 and 4 respectively. It can be easily seen that all metrics indicate that our Positional Integrity Encoding can perfectly maintain the prediction accuracy of full re-computation in different scenarios. This indicates that PIE effectively addresses temporal confusion, and the semantic impact of minor edits is relatively negligible. For example, in the most challenging XF-F setting requiring to handle long-range cross-file context, the maximum relative difference between our PIE and Full-recomputation across different model sizes and code languages is 0.3%/0.15%, 0.66%/0.79%, 1.33%/2.24% for code insertion, deletion, and edition in terms of EM/ES respectively, which is rather negligible. This thorough examination serves as a strong support of the reliability of our PIE approach in real-world code editing scenarios.

**Positional Integrity Encoding significantly reduces computational overhead.** Moreover, we further benchmark the computational costs brought by different approaches. From all results in the above tables, it can be easily seen that our Positional Integrity Encoding can achieve substantial speed-up compared to full re-computation while preserving its performance simultaneously. In particular, for the code edition experiment that requires both insertion and deletion, the averaged reductions of computational overhead induced by our PIE are 87.9%/88.2%, 92.4%/92.8%, 94.8%/95.2% for 1.3B, 6.7B, and 33B models on Python/Java languages respectively. Furthermore, compared to Conflict Fast Encoding which induces minimal costs but largely hurts performance, our PIE only brings negligible overhead, showing its good accuracy-efficiency balance.

In summary, our main results comprehensively demonstrate the superiority of our Positional Integration Encoding for code LLMs towards real-world code editing scenarios, which perfectly preserves the prediction accuracy and significantly addresses the crucial gap in the efficient deployment of LLMs in real-time dynamic scenarios.

Table 3: **Performance comparisons of deletion experiments.** In this task, for each next-line prediction target, we delete several lines of code of its context randomly to simulate real-world scenarios. EM and ES denotes the Exact Match and Edit Similarity score respectively. All results demonstrate that our Positional Integrity Encoding approach brings substantial speed-ups without performance drops.

| | Model | Method | XF-F | | | XF-R | | | IF | | |
|---|---|---|---|---|---|---|---|---|---|---|---|
| | | | EM | ES | Time | EM | ES | Time | EM | ES | Time |
| Python | 1.3B | Full-recomputation | 22.42 | 65.26 | 192ms | 35.41 | 72.96 | 193ms | 28.78 | 69.22 | 193ms |
| | 1.3B | Conflict Fast Encoding | 0.41 | 39.31 | 22ms | 0.59 | 42.02 | 24ms | 1.05 | 42.85 | 22ms |
| | 1.3B | PIE | 22.31 | 65.19 | 29ms | 35.25 | 72.81 | 27ms | 28.8 | 69.16 | 27ms |
| Python | 6.7B | Full-recomputation | 28.95 | 70.11 | 561ms | 40.89 | 76.19 | 564ms | 35.26 | 72.73 | 562ms |
| | 6.7B | Conflict Fast Encoding | 0.51 | 40.24 | 30ms | 0.77 | 42.77 | 31ms | 1.42 | 43.93 | 30ms |
| | 6.7B | PIE | 28.86 | 70.01 | 43ms | 40.77 | 76.15 | 42ms | 35.09 | 72.67 | 42ms |
| Python | 33B | Full-recomputation | 35.75 | 73.46 | 2194ms | 46.0 | 78.9 | 2199ms | 39.75 | 75.12 | 2194ms |
| | 33B | Conflict Fast Encoding | 1.42 | 43.93 | 105ms | 1.55 | 44.22 | 108ms | 2.4 | 45.04 | 108ms |
| | 33B | PIE | 35.09 | 72.67 | 128ms | 45.21 | 78.18 | 118ms | 39.69 | 75.05 | 119ms |
| Java | 1.3B | Full-recomputation | 26.21 | 70.89 | 200ms | 36.77 | 76.31 | 200ms | 45.89 | 78.04 | 198ms |
| | 1.3B | Conflict Fast Encoding | 0.33 | 33.48 | 22ms | 0.55 | 36.24 | 22ms | 1.35 | 38.65 | 22ms |
| | 1.3B | PIE | 26.05 | 70.77 | 28ms | 36.83 | 76.34 | 27ms | 45.74 | 77.91 | 27ms |
| Java | 6.7B | Full-recomputation | 32.51 | 75.56 | 578ms | 41.97 | 79.41 | 578ms | 50.86 | 80.53 | 578ms |
| | 6.7B | Conflict Fast Encoding | 0.49 | 33.6 | 29ms | 0.8 | 36.17 | 29ms | 1.84 | 38.82 | 29ms |
| | 6.7B | PIE | 32.57 | 75.56 | 42ms | 41.83 | 79.39 | 43ms | 50.88 | 80.52 | 42ms |
| Java | 33B | Full-recomputation | 35.05 | 76.93 | 2269ms | 44.95 | 80.87 | 2281ms | 53.23 | 81.76 | 2270ms |
| | 33B | Conflict Fast Encoding | 0.91 | 35.06 | 109ms | 1.01 | 37.97 | 106ms | 2.31 | 40.15 | 105ms |
| | 33B | PIE | 34.82 | 76.76 | 117ms | 44.93 | 80.86 | 120ms | 53.19 | 81.71 | 119ms |

Table 4: **Performance comparisons of edition experiments.** In this task, for each next-line prediction target, we delete several lines of code of its context and simultaneously insert other lines of code randomly to simulate real-world scenarios. EM and ES denote the Exact Match and Edit Similarity score respectively. All results demonstrate that our Positional Integrity Encoding approach brings substantial speed-ups without performance drops.

| | Model | Method | XF-F | | | XF-R | | | IF | | |
|---|---|---|---|---|---|---|---|---|---|---|---|
| | | | EM | ES | Time | EM | ES | Time | EM | ES | Time |
| Python | 1.3B | Full-recomputation | 22.42 | 65.26 | 242ms | 35.41 | 72.96 | 242ms | 28.78 | 69.22 | 244ms |
| | 1.3B | Conflict Fast Encoding | 8.80 | 50.08 | 23ms | 13.83 | 54.01 | 22ms | 11.29 | 51.99 | 22ms |
| | 1.3B | PIE | 22.04 | 64.59 | 30ms | 34.49 | 72.02 | 29ms | 28.15 | 68.32 | 29ms |
| Python | 6.7B | Full-recomputation | 28.95 | 70.11 | 705ms | 40.89 | 76.19 | 706ms | 35.26 | 72.73 | 713ms |
| | 6.7B | Conflict Fast Encoding | 11.26 | 51.63 | 34ms | 12.57 | 53.27 | 34ms | 12.75 | 51.45 | 34ms |
| | 6.7B | PIE | 28.07 | 69.00 | 54ms | 39.89 | 75.10 | 54ms | 34.05 | 71.64 | 54ms |
| Python | 33B | Full-recomputation | 35.75 | 73.46 | 2766ms | 46.00 | 78.90 | 2759ms | 39.75 | 75.12 | 2787ms |
| | 33B | Conflict Fast Encoding | 14.33 | 53.73 | 126ms | 15.12 | 54.59 | 121ms | 13.88 | 51.94 | 127ms |
| | 33B | PIE | 34.62 | 72.47 | 146ms | 44.59 | 77.69 | 142ms | 38.83 | 74.08 | 141ms |
| Java | 1.3B | Full-recomputation | 26.21 | 70.89 | 251ms | 36.77 | 76.31 | 253ms | 45.89 | 78.04 | 249ms |
| | 1.3B | Conflict Fast Encoding | 5.87 | 31.46 | 22ms | 8.01 | 32.74 | 23ms | 10.20 | 34.34 | 23ms |
| | 1.3B | PIE | 25.29 | 68.93 | 30ms | 35.59 | 74.17 | 30ms | 44.51 | 76.06 | 29ms |
| Java | 6.7B | Full-recomputation | 32.51 | 75.56 | 733ms | 41.97 | 79.41 | 736ms | 50.86 | 80.53 | 728ms |
| | 6.7B | Conflict Fast Encoding | 9.28 | 39.32 | 34ms | 11.47 | 39.11 | 35ms | 15.51 | 41.92 | 33ms |
| | 6.7B | PIE | 31.32 | 73.83 | 52ms | 40.89 | 77.65 | 53ms | 49.44 | 78.80 | 53ms |
| Java | 33B | Full-recomputation | 35.05 | 76.93 | 2892ms | 44.95 | 80.87 | 2894ms | 53.23 | 81.76 | 2833ms |
| | 33B | Conflict Fast Encoding | 8.02 | 32.93 | 120ms | 9.67 | 33.29 | 122ms | 12.10 | 34.70 | 122ms |
| | 33B | PIE | 33.72 | 74.69 | 134ms | 43.71 | 78.66 | 138ms | 51.71 | 79.49 | 143ms |

## 4.3 MORE ANALYSIS

In this subsection, we further present detailed analysis to investigate how large is the gap between our Positional Integrity Encoding and full re-computation in terms of context representations and predictions from LLMs, which provide additional insight of our approach.

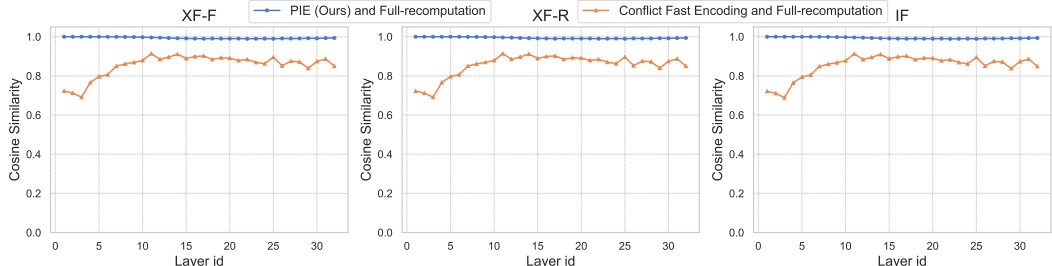

Figure 3: Cosine similarity of key representations across model layers. The plots compare the cosine similarity between $K_{[j+1:n]}$ and $K^*_{[j+1:n]}$ (indicating temporal confusion of Conflict Fast Encoding) with the cosine similarity between $K^{\text{edit}}_{[j+1:n]}$ and $K^*_{[j+1:n]}$ (showing the effectiveness of PIE).

**How large is the gap on context representations?** In practical scenarios, real-time editing by users results in the modified sequence $\mathbf{x}^{\text{edit}}$, requiring the KV cache to must be updated. In our analysis, we use the cosine similarity between context representations of full re-computation $K^*_{[j+1:n]}$ and (1) our Positional Integrity Encoding $K^{\text{edit}}_{[j+1:n]}$; (2) Conflict Fast Encoding $K_{[j+1:n]}$. We employ the DeepSeek-Coder 6.7B model on the Python subset of RepoBench. Averaged results are reported.

In Figure 3, the cosine similarity between representations of full re-computation and our Positional Integraty Enocding is consistently around 1.0 across all layers. This high similarity demonstrates the effectiveness of PIE in preserving the contextual integrity of the key representations after editing, suggesting that PIE successfully mitigates the temporal confusion that typically arises when manipulating the KV cache. In contrast, the cosine similarity between representations of full re-computation and Conflict Fast Encoding is significantly lower, indicating the temporal confusion issue that hurts model performance a lot.

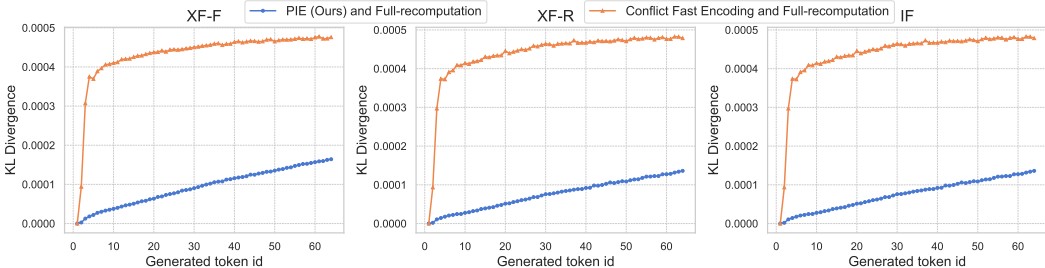

Figure 4: KL divergence of the generated token distributions. The plots compare the KL divergence between the generated token distributions of PIE and Full-recomputation, and the KL divergence between the generated token distributions of Conflict Fast Encoding and Full-recomputation.

**How large is the gap on model predictions?** Moreover, we further use Kullback-Leibler (KL) divergence as the metric to investigate the gap between model predictions of different approaches. Similarly, we employ the DeepSeek-Coder 6.7B model on the Python subset of RepoBench and report the averaged results. In Figure 4, the KL divergence between model predictions of full re-computation and our Positional Integrity Encoding remains consistently low, i.e., below 0.0002 across 64 tokens. However, the KL divergence between model predictions of full re-computation and Conflict Fast Encoding is substantially higher (2x larger). These findings again underscore the importance of maintaining positional integrity within the KV cache to ensure accurate generation results.

## 5 CONCLUSION

In this paper, we introduce Positional Integrity Encoding (PIE), a novel method designed to enhance the efficiency of large language models (LLMs) in the real-time editing setting. Our approach addresses the significant computational overhead associated with re-encoding contexts after small edits, a common scenario in interactive coding environments. Through extensive experiments,

we demonstrated that PIE not only significantly reduces latency but also maintains high accuracy compared to the naive full re-computation method.

PIE represents a substantial step forward in the development of efficient LLMs, particularly in dynamic contexts where frequent edits are made. Future work could explore the integration of PIE with other optimization techniques and its application to a broader range of tasks beyond code generation. Our method paves the way for more responsive and resource-efficient AI assistants, enhancing their practicality and usability in various real-world scenarios.

## ACKNOWLEDGEMENTS

We thank all the anonymous reviewers for the very careful and detailed reviews as well as the valuable suggestions. Their help has further enhanced our work. Di He is supported by National Science Foundation of China (NSFC62376007).

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

## A EXPERIMENTAL DETAILS

### A.1 EXPERIMENTAL SETUP

**Tasks Construction for Code Insertion.** To simulate code insertion tasks, we start by randomly deleting five consecutive lines from each context. The resulting context, which lacks these five lines, is considered the original context. The complete context, which includes the previously deleted lines, is treated as the edited context. The tokens within the deleted lines are identified as the inserted tokens (around 64 tokens for Python and 51 tokens for Java). This setup allows us to evaluate the model's capability to accurately restore missing code segments, mimicking real-world scenarios where developers frequently insert blocks of code.

**Tasks Construction for Code Deletion.** For code deletion tasks, we begin by randomly selecting a line within the context and then inserting five randomly sampled lines at this position. The context containing these additional lines is designated as the original context. The complete context, which excludes the inserted lines, is regarded as the edited context. The tokens in the inserted lines are treated as the deleted tokens (around 64 tokens for Python and 51 tokens for Java). This construction enables us to assess the model's performance in identifying and removing extraneous code, reflecting situations where developers need to clean up or refactor their codebase.

Table 5: **Performance comparisons of insertion experiments for CodeLlama.** In this task, for each next-line prediction target, we insert several lines of code into its context randomly to simulate real-world scenarios. EM and ES denotes the Exact Match and Edit Similarity score respectively. All results demonstrate that our Positional Integrity Encoding approach brings substantial speed-ups without performance drops.

| | Model | Method | XF-F | | | XF-R | | | IF | | |
|---|---|---|---|---|---|---|---|---|---|---|---|
| | | | EM | ES | Time | EM | ES | Time | EM | ES | Time |
| Python | 7B | Full-recomputation | 25.72 | 66.49 | 561ms | 38.87 | 73.81 | 564ms | 33.55 | 70.26 | 562ms |
| | 7B | Conflict Fast Encoding | 14.49 | 55.10 | 34ms | 19.32 | 57.34 | 34ms | 15.49 | 53.05 | 34ms |
| | 7B | PIE | 25.42 | 66.40 | 50ms | 38.76 | 73.83 | 50ms | 33.30 | 70.11 | 50ms |
| Python | 34B | Full-recomputation | 31.04 | 69.48 | 2013ms | 42.80 | 76.27 | 2029ms | 37.69 | 72.36 | 1999ms |
| | 34B | Conflict Fast Encoding | 11.56 | 50.23 | 113ms | 16.06 | 52.59 | 110ms | 11.58 | 48.36 | 112ms |
| | 34B | PIE | 30.53 | 68.79 | 123ms | 42.73 | 76.40 | 119ms | 37.51 | 72.25 | 123ms |
| Java | 7B | Full-recomputation | 28.02 | 72.49 | 578ms | 39.61 | 77.81 | 578ms | 49.20 | 79.76 | 578ms |
| | 7B | Conflict Fast Encoding | 10.45 | 37.69 | 34ms | 14.28 | 38.45 | 35ms | 16.95 | 38.15 | 33ms |
| | 7B | PIE | 28.02 | 72.39 | 50ms | 39.48 | 77.74 | 49ms | 49.31 | 79.79 | 48ms |
| Java | 34B | Full-recomputation | 32.16 | 75.09 | 2179ms | 43.45 | 79.89 | 2184ms | 52.43 | 80.97 | 2197ms |
| | 34B | Conflict Fast Encoding | 12.04 | 41.76 | 104ms | 14.25 | 39.19 | 107ms | 17.29 | 38.50 | 109ms |
| | 34B | PIE | 31.45 | 74.64 | 119ms | 43.54 | 79.80 | 118ms | 52.42 | 81.00 | 115ms |

**Tasks Construction for Multi-place Code Edition**   To comprehensively evaluate the model's performance in handling simultaneous code insertion and deletion, we construct a task scenario that integrates both operations. Initially, we randomly delete five consecutive lines from each context to simulate code insertion. The context without these lines is treated as the original context. The tokens in the deleted lines are identified as the inserted tokens.

Simultaneously, we randomly select another line within the context and insert five randomly sampled lines at this position. The complete context, which includes all lines as they appear after both deletions and insertions, is regarded as the edited context. The tokens in the newly inserted lines are considered the deleted tokens. This dual operation setup allows us to evaluate the model's ability to handle complex, simultaneous edits, adding missing code segments while removing extraneous ones, reflecting the multifaceted nature of real-world coding environments where developers often perform multiple types of edits concurrently.

## A.2    RESULTS ON CODELLAMA

Results of different code editing settings on CodeLlama (Roziere et al., 2023) are presented in Table 5, 6 and 7 respectively. Similar to DeepSeek-Coder, CodeLlama with Positional Integrity Encoding demonstrates strong performance and fast speed across various editing scenarios. The metrics indicate that PIE effectively maintains prediction accuracy and addresses temporal confusion, ensuring the impact of minor edits is minimal.

## B    ADDITIONAL EXPERIMENTS

### B.1    CODE GENERATION TASKS

To further validate the effectiveness of PIE, we conduct experiments on code generation tasks, using HumanEval Chen et al. (2021) and its C++ version from HumanEval-X Zheng et al. (2023). We randomly choose three consecutive lines in the prompt as the user inserted code and report Pass@1 for different methods. It's important to note that the context (prompt) in HumanEval consists of only a few lines (with an average of 15 lines), making each edit semantically significant. The results in Table 8 demonstrate that PIE significantly outperforms the baseline methods, approaching the performance of full recomputation. Note that we also include a reuse baseline where the pre-edit context is reused without modification. Given the importance of these edits in such short contexts, updating the KV cache with PIE becomes essential for maintaining high prediction accuracy, whereas the performance of the reuse baseline drops substantially.

Table 6: **Performance comparisons of deletion experiments for CodeLlama.** In this task, for each next-line prediction target, we delete several lines of code of its context randomly to simulate real-world scenarios. EM and ES denote the Exact Match and Edit Similarity score respectively. All results demonstrate that our Positional Integrity Encoding approach brings substantial speed-ups without performance drops.

| | Model | Method | XF-F | | | XF-R | | | IF | | |
|---|---|---|---|---|---|---|---|---|---|---|---|
| | | | EM | ES | Time | EM | ES | Time | EM | ES | Time |
| Python | 7B | Full-recomputation | 25.72 | 66.49 | 561ms | 38.87 | 73.81 | 564ms | 33.55 | 70.26 | 562ms |
| | 7B | Conflict Fast Encoding | 10.14 | 52.80 | 30ms | 15.24 | 57.88 | 31ms | 13.04 | 54.53 | 30ms |
| | 7B | PIE | 25.52 | 66.47 | 43ms | 38.88 | 73.87 | 42ms | 33.51 | 70.18 | 42ms |
| Python | 34B | Full-recomputation | 31.04 | 69.48 | 2013ms | 42.80 | 76.27 | 2029ms | 37.69 | 72.36 | 1999ms |
| | 34B | Conflict Fast Encoding | 12.15 | 54.76 | 88ms | 16.40 | 59.29 | 94ms | 13.78 | 55.90 | 93ms |
| | 34B | PIE | 31.01 | 69.44 | 100ms | 42.76 | 76.13 | 100ms | 37.60 | 72.38 | 102ms |
| Java | 7B | Full-recomputation | 28.02 | 72.49 | 578ms | 39.61 | 77.81 | 578ms | 49.20 | 79.76 | 578ms |
| | 7B | Conflict Fast Encoding | 7.21 | 42.51 | 29ms | 7.77 | 43.86 | 29ms | 8.78 | 44.46 | 29ms |
| | 7B | PIE | 28.01 | 72.45 | 42ms | 39.51 | 77.78 | 42ms | 49.30 | 79.80 | 42ms |
| Java | 34B | Full-recomputation | 32.16 | 75.09 | 2179ms | 43.45 | 79.89 | 2184ms | 52.43 | 80.97 | 2197ms |
| | 34B | Conflict Fast Encoding | 8.76 | 45.44 | 92ms | 8.67 | 45.57 | 92ms | 10.11 | 46.64 | 95ms |
| | 34B | PIE | 32.12 | 75.02 | 99ms | 43.33 | 79.86 | 98ms | 52.36 | 80.96 | 107ms |

Table 7: **Performance comparisons of edition experiments for CodeLlama.** In this task, for each next-line prediction target, we delete several lines of code of its context and simultaneously insert other lines of code randomly to simulate real-world scenarios. EM and ES denote the Exact Match and Edit Similarity score respectively. All results demonstrate that our Positional Integrity Encoding approach brings substantial speed-ups without performance drops.

| | Model | Method | XF-F | | | XF-R | | | IF | | |
|---|---|---|---|---|---|---|---|---|---|---|---|
| | | | EM | ES | Time | EM | ES | Time | EM | ES | Time |
| Python | 7B | Full-recomputation | 25.72 | 66.49 | 705ms | 38.87 | 73.81 | 706ms | 33.55 | 70.26 | 713ms |
| | 7B | Conflict Fast Encoding | 19.97 | 61.46 | 34ms | 29.79 | 67.31 | 34ms | 24.70 | 63.28 | 34ms |
| | 7B | PIE | 24.97 | 66.04 | 54ms | 38.21 | 73.40 | 54ms | 32.84 | 69.66 | 54ms |
| Python | 34B | Full-recomputation | 31.04 | 69.48 | 2574ms | 42.80 | 76.27 | 2569ms | 37.69 | 72.36 | 2581ms |
| | 34B | Conflict Fast Encoding | 21.91 | 61.78 | 91ms | 30.87 | 67.16 | 91ms | 25.76 | 63.15 | 90ms |
| | 34B | PIE | 29.92 | 68.7 | 126ms | 41.74 | 75.37 | 126ms | 37.01 | 71.76 | 127ms |
| Java | 7B | Full-recomputation | 28.02 | 72.49 | 733ms | 39.61 | 77.81 | 736ms | 49.20 | 79.76 | 728ms |
| | 7B | Conflict Fast Encoding | 18.78 | 57.38 | 34ms | 25.79 | 60.97 | 35ms | 32.60 | 62.66 | 33ms |
| | 7B | PIE | 27.43 | 71.60 | 52ms | 38.89 | 76.96 | 53ms | 48.60 | 78.94 | 53ms |
| Java | 34B | Full-recomputation | 32.16 | 75.09 | 2726ms | 43.45 | 79.89 | 2807ms | 52.43 | 80.97 | 2727ms |
| | 34B | Conflict Fast Encoding | 21.25 | 59.70 | 104ms | 27.83 | 61.91 | 107ms | 32.98 | 62.03 | 109ms |
| | 34B | PIE | 30.82 | 73.84 | 126ms | 42.71 | 78.94 | 119ms | 51.64 | 80.15 | 125ms |

## B.2 CONTEXTUALLY RELATED EDITS

We design a contextually related insertion setting, where we use Edit Distance to identify lines in the context that are similar to the target line, simulating user insertions. We also increase the number of edited lines. We conduct experiments on the Cross-File-First setting of RepoBench-C-8k, keeping all other settings the same as Section 4. The results in Table 9 demonstrate that PIE is robust in handling contextually related edits and outperforms the baseline methods, approaching the performance of full recomputation. Note that we also include a reuse baseline where the pre-edit context is reused without modification.

## B.3 MULTI-PLACE EDITS

Users may edit a function definition followed by updates to all associated call sites. To simulate this scenario, we treat each line in the context as a potential site and use Edit Distance to identify multiple sites in the context that are similar to the target line, thereby simulating multi-place insertions. We conduct experiments in this contextually related multi-place editing setting on the Cross-File-First task of RepoBench-C-8k. The results in Table 10 demonstrate that PIE is robust in handling multi-place edits and outperforms the baseline methods.

Table 8: Performance of DeepSeek-Coder 6.7B on HumanEval and HumanEval-X

| Method | Pass@1 | |
|---|---|---|
| | Python | C++ |
| Full-recomputation | 49.4 | 50.3 |
| Conflict Fast Encoding | 11.0 | 15.5 |
| Reuse baseline | 17.7 | 29.2 |
| PIE | 43.9 | 49.7 |

Table 9: Performance of DeepSeek-Coder 6.7B on contextually-related insertion experiments

| | Model | Method | 5 lines | | 7 lines | | 11 lines | | 13 lines | | 15 lines | |
|---|---|---|---|---|---|---|---|---|---|---|---|---|
| | | | EM | ES | EM | ES | EM | ES | EM | ES | EM | ES |
| Python | 6.7B | Full-recomputation | 28.95 | 70.11 | 28.95 | 70.11 | 28.95 | 70.11 | 28.95 | 70.11 | 28.95 | 70.11 |
| | 6.7B | Conflict Fast Encoding | 7.92 | 41.56 | 6.39 | 38.56 | 4.78 | 34.91 | 4.11 | 33.34 | 3.43 | 32.15 |
| | 6.7B | Reuse baseline | 25.58 | 68.23 | 25.29 | 68.03 | 25.23 | 67.95 | 25.13 | 67.83 | 24.71 | 67.74 |
| | 6.7B | PIE | 27.40 | 69.10 | 27.23 | 68.94 | 27.04 | 68.80 | 26.92 | 68.8 | 26.86 | 68.63 |
| Java | 6.7B | Full-recomputation | 32.51 | 75.56 | 32.51 | 75.56 | 32.51 | 75.56 | 32.51 | 75.56 | 32.51 | 75.56 |
| | 6.7B | Conflict Fast Encoding | 2.81 | 15.46 | 1.89 | 12.16 | 1.10 | 7.89 | 0.87 | 6.52 | 0.74 | 5.57 |
| | 6.7B | Reuse baseline | 25.99 | 73.29 | 25.72 | 73.17 | 25.18 | 73.03 | 25.15 | 72.99 | 24.99 | 72.84 |
| | 6.7B | PIE | 30.04 | 74.46 | 30.00 | 74.44 | 30.13 | 74.47 | 29.87 | 74.38 | 29.76 | 74.29 |

Table 10: Performance of DeepSeek-Coder 6.7B on contextually-related multi-place insertion experiments

| | Model | Method | 5 lines | | 7 lines | | 11 lines | | 13 lines | | 15 lines | |
|---|---|---|---|---|---|---|---|---|---|---|---|---|
| | | | EM | ES | EM | ES | EM | ES | EM | ES | EM | ES |
| Python | 6.7B | Full-recomputation | 28.95 | 70.11 | 28.95 | 70.11 | 28.95 | 70.11 | 28.95 | 70.11 | 28.95 | 70.11 |
| | 6.7B | Conflict Fast Encoding | 7.43 | 39.95 | 5.83 | 37.00 | 4.06 | 33.02 | 3.48 | 31.71 | 2.97 | 30.59 |
| | 6.7B | Reuse baseline | 24.61 | 67.43 | 24.44 | 67.17 | 23.84 | 66.77 | 23.62 | 66.63 | 23.49 | 66.55 |
| | 6.7B | PIE | 26.97 | 68.85 | 26.77 | 68.67 | 26.33 | 68.34 | 26.15 | 68.222 | 25.9 | 68.23 |
| Java | 6.7B | Full-recomputation | 32.51 | 75.56 | 32.51 | 75.56 | 32.51 | 75.56 | 32.51 | 75.56 | 32.51 | 75.56 |
| | 6.7B | Conflict Fast Encoding | 1.69 | 10.91 | 1.06 | 7.66 | 0.43 | 4.26 | 0.28 | 3.36 | 0.22 | 2.81 |
| | 6.7B | Reuse baseline | 24.46 | 72.35 | 23.99 | 72.05 | 23.11 | 71.69 | 22.97 | 71.56 | 22.75 | 71.38 |
| | 6.7B | PIE | 29.56 | 74.14 | 29.26 | 73.91 | 29.03 | 73.81 | 28.83 | 73.68 | 28.56 | 73.59 |

