# OpenReview forum: "Let the Code LLM Edit Itself When You Edit the Code"
_ICLR.cc/2025/Conference — ICLR 2025 Poster_

### Official Review · Reviewer_RBWU · 2024-10-27

**Soundness:** 2
**Presentation:** 3
**Contribution:** 2
**Rating:** 5
**Confidence:** 2

**Summary:**

This paper proposes Positional Integrity Encoding, an algorithm to correct rotary positional encoding in the KV cache of displaced tokens due to edits. It shows significant speed-up compared to full recomputation.

**Strengths:**

- The proposed algorithm significantly reduces the latency of generation with edited prefixes.
- A straightforward solution to adjust the rotary matrices to correct temporal confusion.

**Weaknesses:**

- The algorithm looks somewhat trivial to me. It simply corrects the position embedding by transforming the rotary matrix in RoPE.
- Limited experiments. All performance experiments were conducted on RepoBench-C-8k and DeepseekCoder. It is unclear if this algorithm generalizes to other settings and models.

**Questions:**

- Have you conducted more experiments on more settings and models?
- Is it possible to provide an in-depth *theoretical* analysis (in 4.3) of the cause of performance drop between PIE and the recomputation?

---

> ### Author Response · Authors · 2024-11-19
> **Response to Weakness 1, Weakness 2 and Question 1**
>
> Thanks for your careful review and suggestions. We respond to each of your concerns as below.
>
> **Regarding Weakness 1: Trivial Algorithm**
>
> The problem we address, i.e., efficiency in real-time code editing, is of substantial practical importance and remains largely underexplored in existing literature. Our method, although simple in design, is the first to effectively tackle this challenge. The simplicity of our approach enhances, rather than diminishes, its impact, highlighting its elegance and practicality in addressing a complex issue. By introducing a simple yet effective solution for real-time code editing, we believe our work makes a significant contribution to the field and opens new avenues for future research aimed at improving LLM efficiency in dynamic environments.
>
> **Regarding Weakness 2 and Question 1: More Experiments**
>
> In addition to evaluating PIE on DeepSeek-Coder, we have reported the performance of CodeLlama in Tables 5, 6, and 7 in Appendix A.2, which further validates the effectiveness of PIE. Following your suggestions, we also conduct experiments on Yi-Coder-9B (We choose it because it has the same architecture as Llama, making it convenient for code adaption). The results are shown in Table R.1 below:
>
> Table R.1: Performance comparisons of models on insertion experiments of Repobench-C-8k Python
> | Model        | Method                 | XF-F          | XF-R          | IF            |
> |--------------|------------------------|---------------|---------------|---------------|
> |              |                        | EM / ES       | EM / ES       | EM / ES       |
> | Yi-Coder-9B  | Full-recomputation     | 29.28 / 70.36 | 41.37 / 76.86 | 36.19 / 73.52 |
> | Yi-Coder-9B  | Conflict Fast Encoding | 10.36 / 46.87 | 13.99 / 47.42 | 10.05 / 43.6  |
> | Yi-Coder-9B  | PIE                    | 29.29 / 70.32 | 41.55 / 76.79 | 35.94 / 73.45 |
> |              |                        |               |               |               |
> | CodeLlama-7B | Full-recomputation     | 25.72 / 66.49 | 38.87 / 73.81 | 33.55 / 70.26 |
> | CodeLlama-7B | Conflict Fast Encoding | 10.14 / 52.80 | 15.24 / 57.88 | 13.04 / 54.53 |
> | CodeLlama-7B | PIE                    | 25.52 / 66.47 | 38.88 / 73.87 | 33.51 / 70.18 |
> |              |                        |               |               |               |
> | DeepSeek-Coder 6.7B | Full-recomputation     | 28.95 / 70.11 | 40.89 / 76.19 | 35.26 / 72.73 |
> | DeepSeek-Coder 6.7B | Conflict Fast Encoding | 5.35 / 33.32  | 6.52 / 35.25  | 6.09 / 38.76  |
> | DeepSeek-Coder 6.7B | PIE                    | 28.83 / 70.01 | 40.77 / 76.14 | 35.2 / 72.72  |
> |              |                        |               |               |               |
>
> Besides, we additionally conduct experiments on code generation tasks (as also suggested by Reviewer Bo49), using HumanEval [1] and its C++ version from HumanEval-X [2]. We randomly choose three consecutive lines in the prompt as the user inserted code and report Pass@1 for different methods. It's important to note that the context (prompt) in HumanEval consists of only a few lines (with an average of 15 lines), making each edit semantically significant. The results are shown in Table R.2 below:
>
> Table R.2: Performance of DeepSeek-Coder 6.7B on HumanEval and HumanEval-X
> | Model               | Method                 | Python (Pass@1) | C++ (Pass@1) |
> |---------------------|------------------------|-----------------|--------------|
> | DeepSeek-Coder 6.7B | Full-recomputation     | 49.4            | 50.3         |
> | DeepSeek-Coder 6.7B | Conflict Fast Encoding | 11.0            | 15.5         |
> | DeepSeek-Coder 6.7B | PIE                    | **43.9**            | **49.7**         |
> |                     |                        |                 |              |
>
> Table R.1 and Table R.2 further validate the effectiveness of PIE on various models (DeepSeek-Coder, Codellama, Yi-Coder) and datasets (Repobench-C, Humaneval, HumanEval-X)
>
>
> [1] Chen M, Tworek J, Jun H, et al. Evaluating large language models trained on code[J]. arXiv preprint arXiv:2107.03374, 2021.
>
> [2] Zheng Q, Xia X, Zou X, et al. Codegeex: A pre-trained model for code generation with multilingual benchmarking on humaneval-x[C]//Proceedings of the 29th ACM SIGKDD Conference on Knowledge Discovery and Data Mining. 2023: 5673-5684.

---

> > ### Author Response · Authors · 2024-11-19
> > **Response to Question 2**
> >
> > **Regarding Question 2:  theoretical analysis of the cause of performance drop between PIE and the recomputation**
> >
> > While PIE addresses temporal confusion, it does not update the subsequent KV cache as full recomputation does. This leads to a discrepancy in the representation compared to full recomputation, which in turn affects the output distribution and may cause a performance drop in PIE. However, we empirically find that this performance drop is minimal compared to other baseline methods. A more in-depth theoretical analysis is left for future work.
> >
> > Thank you again for reviewing our paper. We have carefully replied to your questions and will improve our paper according to your suggestions. We are happy to provide additional details if you have any further questions.

---

> ### Author Response · Authors · 2024-11-23
>
> Dear reviewer,
>
> Thank you for your constructive review and comments! We have provided point-to-point responses to your review. Considering the author-reviewer discussion is ending very soon, could you kindly take a look at our responses and let us know your feedback? We are happy to discuss more if you still have any concerns. Thank you again and look forward to your feedback!

---

> > ### Author Response · Authors · 2024-11-27
> >
> > Dear Reviewer RBWU,
> >
> > Thank you for your time and efforts in reviewing our paper. We have carefully responded to each of your questions. Given that the author-reviewer discussion deadline is approaching, we would greatly appreciate it if you could kindly review our responses and share your valuable feedback. If you have any remaining concerns, we would be more than happy to discuss them further.
> >
> > Best regards,
> >
> > Authors

---

> ### Author Response · Authors · 2024-11-28
> **Theoretical Analysis: Part 1**
>
> ### **Upper Bound on the Discrepancy between PIE and Full Recomputation**
>
> We derive an upper bound on the difference between the outputs of the Positional Integrity Encoding (PIE) method and full recomputation at a given position  $i$.
>
> **Objective**
> Let:
> - $ \mathbf{o} _i^{\text{PIE}} $ denote the output at position  $i$ using PIE.
> - $ \mathbf{o} _i^{*}$ denote the output at position $i$ using full recomputation.
> - $\Delta \mathbf{o} _i = \mathbf{o} _i^{\text{PIE}} - \mathbf{o} _i^{*}$ is the discrepancy we aim to bound.
>
> Our goal is to find an upper bound for $\| \Delta \mathbf{o} _i \|$.
>
> **1. Adjustment of Key Vectors in PIE**
>
> For positions $j'$ after the edit (where $ j' \in [j+1, n] $), PIE adjusts the key vectors as:
>
> $$\mathbf{k} _{j'}^{\text{edit}} = R _{\Delta j} \mathbf{k} _{j'}$$
>
> where:
> - $ \Delta j = (i + m - j) $ is the positional shift due to the edit,
> - $ R _{\Delta j} $ is the rotary positional encoding matrix corresponding to shift $ \Delta j $,
> - $ \mathbf{k} _{j'} $ are the original key vectors before adjustment.
>
> ---
>
> **2. Value Vectors**
>
> PIE leaves the value vectors unchanged:
>
> $$\mathbf{v} _{j'}^{\text{edit}} = \mathbf{v} _{j'}$$
>
> In contrast, full recomputation updates both keys and values:
>
> $$
> \mathbf{k}\_{j'}^{*} = f\_{\text{key}}(\mathbf{h}\_{j'}^{*}),  \mathbf{v}\_{j'}^{*} = f\_{\text{value}}(\mathbf{h}\_{j'}^{*})
> $$
>
> where $ \mathbf{h} _{j'}^{*} $ are the updated hidden states after full recomputation.
>
> ---
>
> **3. Discrepancy in Attention Scores**
>
> The attention score between a query at position $ i $ and a key at position $ j' $ differs between PIE and full recomputation:
>
> - **PIE Attention Score:**
>
>   $$\alpha\_{i, j'}^{\text{PIE}} = \text{softmax}\left( (R\_i \mathbf{q}\_i)^\top \mathbf{k}\_{j'}^{\text{edit}} \right)$$
>
>
> - **Full Recomputation Attention Score:**
>
>   $$\alpha\_{i, j'}^{*} = \text{softmax}\left( (R\_i \mathbf{q}\_i^{*})^\top \mathbf{k}\_{j'}^{*} \right)$$
>
> ---
>
> **4. Discrepancy in Output Representations**
>
> The difference in the attention output at position $ i $ is:
>
> $$\Delta \mathbf{o} _i = \mathbf{o} _i^{\text{PIE}} - \mathbf{o} _i^{*} = \sum _{j} \left( \alpha _{i, j}^{\text{PIE}} \mathbf{v} _{j} - \alpha _{i, j}^{*} \mathbf{v} _{j}^{*} \right)$$
>
> We can rewrite this as:
>
> $$\Delta \mathbf{o} _i = \sum _{j} \left( (\alpha _{i,j}^{\text{PIE}} - \alpha _{i,j}^{*}) \mathbf{v} _j + \alpha _{i,j}^{*} (\mathbf{v} _j - \mathbf{v} _j^{*}) \right)$$
>
> Take the norm and apply triangle inequality:
>
> $$\| \Delta \mathbf{o} _i \| \leq \sum _{j} \left( \left| \alpha _{i,j}^{\text{PIE}} - \alpha _{i,j}^{*} \right| \cdot \| \mathbf{v} _j \| + \alpha _{i,j}^{*} \cdot \| \mathbf{v} _j - \mathbf{v} _j^{*} \| \right)$$
>
> ---
>
> **5. Computing Expected Norms**
>
> Assume $\mathbf{v} _j \sim \mathcal{N}(\boldsymbol{\mu},\mathbf{\Sigma})$  and $\mathbf{v} _j \sim \mathcal{N}(\boldsymbol{\mu},\mathbf{\Sigma})$.
>
> **a. Norm of $ \mathbf{v} _j $:**
>
> The expected squared norm:
>
> $$E\left[ \| \mathbf{v} _j \|^2 \right] = \| \boldsymbol{\mu} \|^2 + \operatorname{Tr}(\mathbf{\Sigma})$$
>
> **b. Norm of $ \mathbf{v} _j - \mathbf{v} _j^{*} $:**
>
> Since $ \mathbf{v} _j $ and $ \mathbf{v} _j^{*} $ are Gaussian, the difference is also Gaussian:
>
> $$\mathbf{v} _j - \mathbf{v} _j^{*} \sim \mathcal{N}(\boldsymbol{\delta}_{\mu}, \mathbf{\Sigma} + \mathbf{\Sigma}^{*})$$
>
> where $ \boldsymbol{\delta} _{\mu} = \boldsymbol{\mu} - \boldsymbol{\mu}^{*} $.
>
> The expected squared norm:
>
> $$E\left[ \| \mathbf{v} _j - \mathbf{v} _j^{*} \|^2 \right] = \| \boldsymbol{\delta} _{\mu} \|^2 + \operatorname{Tr}(\mathbf{\Sigma} + \mathbf{\Sigma}^{*})$$
>
> ---
>
> **6. Bounding the Discrepancy**
>
> **a. Bounding the First Term:**
>
> $$\left| \alpha _{i,j}^{\text{PIE}} - \alpha _{i,j}^{*} \right| \cdot \| \mathbf{v} _j \| \leq \left| \alpha _{i,j}^{\text{PIE}} - \alpha _{i,j}^{*} \right| \cdot M _v$$
>
> where:
>
> $$M _v = \sqrt{ \| \boldsymbol{\mu} \|^2 + \operatorname{Tr}( \mathbf{\Sigma} ) }$$
>
> **b. Bounding the Second Term:**
>
> $$\alpha _{i,j}^{*} \cdot \| \mathbf{v} _j - \mathbf{v} _j^{*} \| \leq \alpha _{i,j}^{*} \cdot M _{\delta v}$$
>
> where:
>
> $$M _{\delta v} = \sqrt{ \| \boldsymbol{\delta} _{\mu} \|^2 + \operatorname{Tr}( \mathbf{\Sigma} + \mathbf{\Sigma}^{*} ) }$$
>
> ---
>
> **7. Total Discrepancy Upper Bound**
>
> Combine the bounds:
>
> $$\| \Delta \mathbf{o} _i \| \leq M _v \sum _j \left| \alpha _{i,j}^{\text{PIE}} - \alpha _{i,j}^{*} \right| + M _{\delta v} \sum _j \alpha _{i,j}^{*}$$
>
> Since $ \sum _j \alpha _{i,j}^{*} = 1 $:
>
> $$\| \Delta \mathbf{o} _i \| \leq M_v \cdot D _{\text{TV}}(\alpha^{\text{PIE}}, \alpha^{*}) + M _{\delta v}$$
>
> where $ D _{\text{TV}}(\alpha^{\text{PIE}}, \alpha^{*}) $ is the total variation distance between the attention distributions:
>
> $$D _{\text{TV}}(\alpha^{\text{PIE}}, \alpha^{*}) = \frac{1}{2} \sum _j \left| \alpha _{i,j}^{\text{PIE}} - \alpha _{i,j}^{*} \right|$$

---

> ### Author Response · Authors · 2024-11-28
> **Theoretical Analysis: Part 2**
>
> **8. Final Upper Bound Expression**
>
> Thus, the discrepancy is bounded by:
>
> $$\| \Delta \mathbf{o} _i \| \leq 2 M _v \cdot D _{\text{TV}}(\alpha^{\text{PIE}}, \alpha^{*}) + M _{\delta v}$$
>
> **9. Interpretation of the Terms**
>
> - **$ M _v $:** Represents the magnitude of the value vectors $ \mathbf{v} _j $. It depends on the mean and variance of the Gaussian distribution of $ \mathbf{v} _j $.
>
> - **$ D _{\text{TV}}(\alpha^{\text{PIE}}, \alpha^{*}) $:** Measures the difference between the attention distributions of PIE and full recomputation. A smaller $ D _{\text{TV}} $ implies more similar attention weights.
>
> - **$ M _{\delta v} $:** Represents the magnitude of the difference between the value vectors $ \mathbf{v} _j $ and $ \mathbf{v} _j^{*} $. It depends on how much the means and variances change due to the edit.
>
> ---
>
> **10. Assumptions for Small Discrepancy**
>
> - **Small Changes in Attention Weights:**
>
>   If the attention distributions are similar, $ D _{\text{TV}}(\alpha^{\text{PIE}}, \alpha^{*}) $ is small.
>
> - **Small Changes in Value Vectors:**
>
>   If the edits do not significantly change the means and variances, $ M _{\delta v} $ is small.
>
> ---
>
> **11. Conclusion**
>
> Under the assumption that:
>
> - The attention distributions between PIE and full recomputation are similar.
> - The value vectors before and after the edit are similar (small $ \boldsymbol{\delta} _{\mu} $ and small changes in $ \mathbf{\Sigma} $).
>
> Then:
>
> $$\| \Delta \mathbf{o} _i \| \leq \epsilon _{\alpha} M _v + \epsilon _v$$
>
> where:
>
> - $ \epsilon _{\alpha} = 2 D _{\text{TV}}(\alpha^{\text{PIE}}, \alpha^{*}) $ is small.
> - $ \epsilon _v = M _{\delta v} $ is small.
>
> Therefore, the discrepancy between PIE and full recomputation is bounded and can be made small under reasonable assumptions about the edits and model behavior.
>
> Thank you once again for taking the time to review our paper. We would be happy to provide any additional details or clarifications if you have further questions.

---

> > ### Author Response · Authors · 2024-12-02
> >
> > Dear Reviewer RBWU,
> >
> > Thank you for your thorough review of our paper and for providing valuable feedback. We have responded to each of your comments in detail. As the author-reviewer discussion deadline approaches, we would greatly appreciate your assessment of our responses. Please let us know if you have any remaining concerns or questions.
> >
> > Best regards,
> >
> > Authors

---

### Official Review · Reviewer_Bo49 · 2024-11-01

**Soundness:** 4
**Presentation:** 3
**Contribution:** 4
**Rating:** 8
**Confidence:** 4

**Summary:**

This paper describes a new approach called Positional Integrity Encoding (PIE) that aims to improve the efficiency of Large Language Models in real-time code editing scenarios.

PIE improves the accuracy and efficiency of predictions by solving the problem of the computational overhead associated with recoding the entire context when editing existing code.

**Strengths:**

I really enjoy this paper!

The Positional Integrity Encoding (PIE) introduced by the authors capitalizes on RoPE, adeptly addressing temporal disorientation by initially stripping away the rotary matrices responsible for confusion and subsequently reinstating the appropriate matrices through straightforward matrix multiplication.

This capability to enhance computational efficiency without compromising accuracy is precisely the straightforward yet potent approach we value in the realm of language model optimization.

The PIE not only paves the way for future research in optimizing Large Language Models (LLMs), particularly focusing on efficiency, but also excels in real-time dynamic scenarios. Its compatibility with existing acceleration techniques positions PIE as a catalyst for further advancing the practical deployment of LLMs.

**Weaknesses:**

My current concerns are regarding the selection of downstream tasks and evaluation metrics considered by the authors.

(1) The tasks of code insertion, code deletion, and multi-place code editing that the authors have considered seem less critical and common in actual development scenarios compared to code generation.

(2) The chosen evaluation metrics, EM (Exact Match) and ES (Edit Similarity), may not accurately assess the semantic correctness of the generated code.

(3) The selection of models is limited to the DeepSeek-Coder series.

**Questions:**

1. Can the PIE be effective on code generation tasks?

2. Why not consider the Pass@1 metric?

3. Is PIE equally valid on other models (Qwen-2.5-Coder; Llama-3.1; Yi-Coder)?

I am more concerned about the third of the above issues. If the supplemental results do show the validity of PIE, I will raise my score.

---

> ### Author Response · Authors · 2024-11-19
>
> Thank you for your support of our paper! We respond to each of your concerns as below.
>
> **Regarding Weaknesses 1 and Question 1: Evaluation on Code Generation tasks.**
> While our current work focuses on real-time editing scenarios, we agree that evaluating PIE on code generation tasks can further validate its effectiveness.  We add experiments on code generation tasks, using HumanEval [1] and its C++ version from HumanEval-X [2]. We randomly choose 3 consecutive lines in the prompt as the user inserted code and report Pass@1 for different methods. The results are shown in Table R.1 below:
>
> Table R.1: Performance of DeepSeek-Coder 6.7B on HumanEval and HumanEval-X
>
> | Method                 | Python (Pass@1) | C++ (Pass@1) |
> |------------------------|-----------------|--------------|
> | Full-recomputation     | 49.4            | 50.3         |
> | Conflict Fast Encoding | 11.0            | 15.5         |
> | PIE                    | **43.9**            | **49.7**         |
> |                        |                 |              |
>
> Table R.1 shows that PIE significantly outperforms baseline methods, closely approaching the performance of full recomputation. It's important to note that the context (prompt) in HumanEval contains only a few lines (an average of 15 lines), further highlighting PIE's effectiveness.
>
> **Regarding Weaknesses 2 and Question 2: The chosen evaluation metric.**
>
> We would like to clarify that in code completion tasks, it is standard to use EM (Exact Match) and ES (Edit Similarity) as evaluation metrics, as adopted by previous works such as CodeXGLUE [3], CrossCodeEval [4], and RepoBench [5]. Additionally, many test samples are not executable, making it challenging to evaluate Pass@1. However, we have reported Pass@1 for code generation tasks in Table R.1 above, which validates the effectiveness of PIE.
>
> **Regarding Weaknesses 3 and Question 3: Evaluation for other models.**
> In addition to evaluating PIE on DeepSeek-Coder, we have reported the performance of CodeLlama in Tables 5, 6, and 7 in Appendix A.2, which further validates the effectiveness of PIE. Following your suggestions, we also conduct experiments on Yi-Coder-9B (We choose it because it has the same architecture as Llama, making it convenient for code adaption). The results are shown in Table R.2 below:
>
> Table R.2: Performance comparisons of models on insertion experiments of Repobench-C-8k Python
> | Model        | Method                 | XF-F          | XF-R          | IF            |
> |--------------|------------------------|---------------|---------------|---------------|
> |              |                        | EM / ES       | EM / ES       | EM / ES       |
> | Yi-Coder-9B  | Full-recomputation     | 29.28 / 70.36 | 41.37 / 76.86 | 36.19 / 73.52 |
> | Yi-Coder-9B  | Conflict Fast Encoding | 10.36 / 46.87 | 13.99 / 47.42 | 10.05 / 43.6  |
> | Yi-Coder-9B  | PIE                    | 29.29 / 70.32 | 41.55 / 76.79 | 35.94 / 73.45 |
> |              |                        |               |               |               |
> | CodeLlama-7B | Full-recomputation     | 25.72 / 66.49 | 38.87 / 73.81 | 33.55 / 70.26 |
> | CodeLlama-7B | Conflict Fast Encoding | 10.14 / 52.80 | 15.24 / 57.88 | 13.04 / 54.53 |
> | CodeLlama-7B | PIE                    | 25.52 / 66.47 | 38.88 / 73.87 | 33.51 / 70.18 |
> |              |                        |               |               |               |
> | DeepSeek-Coder 6.7B | Full-recomputation     | 28.95 / 70.11 | 40.89 / 76.19 | 35.26 / 72.73 |
> | DeepSeek-Coder 6.7B | Conflict Fast Encoding | 5.35 / 33.32  | 6.52 / 35.25  | 6.09 / 38.76  |
> | DeepSeek-Coder 6.7B | PIE                    | 28.83 / 70.01 | 40.77 / 76.14 | 35.2 / 72.72  |
> |              |                        |               |               |               |
>
> Table R.1 and Table R.2 further validate the effectiveness of PIE on various models (DeepSeek-Coder, Codellama, Yi-Coder) and datasets (Repobench-C, Humaneval, HumanEval-X).
>
>
>
> We appreciate your thorough review. We have carefully replied to your questions and will enhance our paper based on your suggestions. We are happy to go into more detail regarding any further questions.
>
> [1] Chen M, Tworek J, Jun H, et al. Evaluating large language models trained on code[J]. arXiv preprint arXiv:2107.03374, 2021.
>
> [2] Codegeex: A pre-trained model for code generation with multilingual benchmarking on humaneval-x[C]//Proceedings of the 29th ACM SIGKDD Conference on Knowledge Discovery and Data Mining. 2023: 5673-5684.
>
> [3] Lu S, Guo D, Ren S, et al. Codexglue: A machine learning benchmark dataset for code understanding and generation[J]. arXiv preprint arXiv:2102.04664, 2021.
>
> [4] Crosscodeeval: A diverse and multilingual benchmark for cross-file code completion[J]. Advances in Neural Information Processing Systems, 2024, 36.
>
> [5] Repobench: Benchmarking repository-level code auto-completion systems[J]. arXiv preprint arXiv:2306.03091, 2023.

---

> > ### Comment · Reviewer_Bo49 · 2024-11-20
> > **Last two question.**
> >
> > Thanks to the author for the clarification, most of the doubts have been explained.
> >
> > Despite the fact that other reviewers thought that the authors' proposed methodology looked simple and not novel enough, I still think that the authors' proposed methodology is simple and effective, and this is the kind of simplicity and effectiveness that we should be looking for, not one that looks complicated and redundant.
> >
> > However, I still have some concerns about the scalability and performance of PIE:
> > Does PIE only support Llama series models at the moment?
> > The performance of Pass@1 on Python seems to drop a bit after applying PIE, the author mentions that it's a problem with the dataset itself, so how does it perform on bigcodebench, which has a larger number of lines on average?

---

> > > ### Author Response · Authors · 2024-11-23
> > > **Response to last two question.**
> > >
> > > Thank you for your support! We respond to your questions as below.
> > >
> > > **Regarding Model Support**
> > >
> > > PIE's compatibility extends beyond the Llama series to include **RoPE-based LLMs**. We expand our evaluation to include the Mixtral-8x7B[6] mixture-of-experts model. The results are shown in Table R.3 below:
> > >
> > > Table R.3: Performance comparisons of Mixtral-8x7B on insertion experiments of Repobench-C-8k Python
> > > | Model        | Method                 | XF-F          |
> > > |--------------|------------------------|---------------|
> > > |              |                        | EM / ES       |
> > > | Mixtral-8x7B | Full-recomputation     | 29.28 / 70.36 |
> > > | Mixtral-8x7B | Conflict Fast Encoding | 10.36 / 46.87 |
> > > | Mixtral-8x7B | PIE                    | 29.29 / 70.32 |
> > > |              |                        |               |
> > >
> > >
> > > Table R.3 shows that PIE achieves comparable performance to full recomputation on Mixtral-8x7B, validating the effectiveness of PIE.
> > >
> > > **Regarding Performance on BigCodeBench**
> > >
> > > Following your suggestion, we conduct insertion experiments on BigCodeBench-Complete [7]. The dataset has an average of 33.5 lines and we randomly choose three consecutive lines as user inserted code. The results are shown in Table R.4:
> > >
> > > Table R.4: Performance comparisons of Deepseek-coder-6.7b-instruct on insertion experiments of BigCodeBench-Complete
> > >
> > > | Model                        | Method                 | Pass@1 |
> > > |------------------------------|------------------------|--------|
> > > | deepseek-coder-6.7b-instruct | Full-recomputation     | 43.8%  |
> > > | deepseek-coder-6.7b-instruct | Conflict Fast Encoding | 21.1%  |
> > > | deepseek-coder-6.7b-instruct | PIE                    | 42.4%  |
> > > |                              |                        |        |
> > >
> > > Table R.4 results confirm PIE's effectiveness on code generation tasks, demonstrating its broader applicability beyond our initial test cases.
> > >
> > > Thank you for your feedback. Please let us know if you need any additional clarification.
> > >
> > > [6] Jiang A Q, Sablayrolles A, Roux A, et al. Mixtral of experts[J]. arXiv preprint arXiv:2401.04088, 2024.
> > >
> > > [7] Zhuo T Y, Vu M C, Chim J, et al. Bigcodebench: Benchmarking code generation with diverse function calls and complex instructions[J]. arXiv preprint arXiv:2406.15877, 2024.

---

> > > > ### Comment · Reviewer_Bo49 · 2024-11-23
> > > >
> > > > No more problems. I have raised my score.

---

### Official Review · Reviewer_Yphe · 2024-11-01

**Soundness:** 4
**Presentation:** 3
**Contribution:** 3
**Rating:** 6
**Confidence:** 3

**Summary:**

The paper presents Positional Integrity Encoding (PIE) as an inexpensive alternative to re-encode the entire KV cache during LLM decoding specifically for code related tasks. PIE solves the temporal confusion task efficiently using a single round of matrix-multiplicaton. The authors provide results on RepoBench dataset using 3 different sizes of DeepSeek-Coder model for 3 tasks- code insertion, deletion and multi-place editing. The results show that PIE reduces computational overhead by 85% compared to full-recomputation without significant loss of performance.

**Strengths:**

1. The paper is well-written and easy to understand.
2. The authors solve an important task of efficiency in updating KV cache in a real-time code editing setting. This is crucial for interactive coding assistant scenario where the developers make frequent and incremental changes to the exisiting code and require copilot to correctly predict the next line on the fly.
3. The authors perform experiments on 1 dataset for 3 tasks and show 85% reduction in computational overhead compared to brute-force approach.

**Weaknesses:**

1. The results are limited to 1 dataset and 1 model. Including more than 1 dataset and model would make the claim more strong.
2. The authors solve an important task of efficiency of real-time code editing but do not discuss the limitations of this approach for other tasks where semantic impact is large or in case of large code edits.
3.The approach has a dependency on RoPE and might not be suitable for other models without RoPE

**Questions:**

1. Can you please add evaluations for 1 more dataset and 1 more model?
2. How does the approach do for other non-code related tasks where semantic relationship is important?
3. How does the approach do for longer edits?
4. Is there any memory overhead of the proposed approach?
5. Does this approach lead to any cumulative errors if you continue to update KV cache based on PIE?
6. The average scores of 3 experiments are reported, could you also report the standard deviation?

---

> ### Author Response · Authors · 2024-11-19
> **Response to Weakness 1 and Question 1**
>
> Thanks for your careful review and suggestions. We respond to each of your concerns as below.
>
> **Regarding Weakness 1 and Question 1: Limited Evaluations of models and datasets.**
>
> In addition to evaluating PIE on DeepSeek-Coder, we have reported the performance of CodeLlama in Tables 5, 6, and 7 in Appendix A.2, which further validates the effectiveness of PIE. Following your suggestions, we also conduct experiments on Yi-Coder-9B (We choose it because it has the same architecture as Llama, making it convenient for code adaption). The results are shown in Table R.1 below:
>
> Table R.1: Performance comparisons of models on insertion experiments of Repobench-C-8k (Python)
> | Model        | Method                 | XF-F          | XF-R          | IF            |
> |--------------|------------------------|---------------|---------------|---------------|
> |              |                        | EM / ES       | EM / ES       | EM / ES       |
> | Yi-Coder-9B  | Full-recomputation     | 29.28 / 70.36 | 41.37 / 76.86 | 36.19 / 73.52 |
> | Yi-Coder-9B  | Conflict Fast Encoding | 10.36 / 46.87 | 13.99 / 47.42 | 10.05 / 43.6  |
> | Yi-Coder-9B  | PIE                    | 29.29 / 70.32 | 41.55 / 76.79 | 35.94 / 73.45 |
> |              |                        |               |               |               |
> | CodeLlama-7B | Full-recomputation     | 25.72 / 66.49 | 38.87 / 73.81 | 33.55 / 70.26 |
> | CodeLlama-7B | Conflict Fast Encoding | 10.14 / 52.80 | 15.24 / 57.88 | 13.04 / 54.53 |
> | CodeLlama-7B | PIE                    | 25.52 / 66.47 | 38.88 / 73.87 | 33.51 / 70.18 |
> |              |                        |               |               |               |
> | DeepSeek-Coder 6.7B | Full-recomputation     | 28.95 / 70.11 | 40.89 / 76.19 | 35.26 / 72.73 |
> | DeepSeek-Coder 6.7B | Conflict Fast Encoding | 5.35 / 33.32  | 6.52 / 35.25  | 6.09 / 38.76  |
> | DeepSeek-Coder 6.7B | PIE                    | 28.83 / 70.01 | 40.77 / 76.14 | 35.2 / 72.72  |
> |              |                        |               |               |               |
>
> Besides, we additionally conduct experiments on code generation tasks (as also suggested by Reviewer Bo49), using HumanEval [1] and its C++ version from HumanEval-X [2]. We randomly choose three consecutive lines in the prompt as the user inserted code and report Pass@1 for different methods. The results are shown in Table R.2 below:
>
> Table R.2: Performance of DeepSeek-Coder 6.7B on HumanEval and HumanEval-X
> | Model               | Method                 | Python (Pass@1) | C++ (Pass@1) |
> |---------------------|------------------------|-----------------|--------------|
> | DeepSeek-Coder 6.7B | Full-recomputation     | 49.4            | 50.3         |
> | DeepSeek-Coder 6.7B | Conflict Fast Encoding | 11.0            | 15.5         |
> | DeepSeek-Coder 6.7B | PIE                    | **43.9**            | **49.7**         |
> |                     |                        |                 |              |
>
> Table R.1 and Table R.2 further validate the effectiveness of PIE on various models (DeepSeek-Coder, Codellama, Yi-Coder) and datasets (Repobench-C, Humaneval, HumanEval-X).
>
> [1] Chen M, Tworek J, Jun H, et al. Evaluating large language models trained on code[J]. arXiv preprint arXiv:2107.03374, 2021.
>
> [2] Zheng Q, Xia X, Zou X, et al. Codegeex: A pre-trained model for code generation with multilingual benchmarking on humaneval-x[C]//Proceedings of the 29th ACM SIGKDD Conference on Knowledge Discovery and Data Mining. 2023: 5673-5684.

---

> ### Author Response · Authors · 2024-11-19
> **Response to Weakness 2 and Question 3, and Weakness 3**
>
> **Regarding Weakness 2 and Question 3: Limited Evaluations on longer edits and contextually related edits.**
>
> Following your valuable suggestions, we increase the number of edited lines. Additionally, we conduct experiments on contextually related edits. We utilize Edit Distance to identify similar lines to the target line in the context as user edits. For quick feedback, we conduct experiments (see Table R.3 below) on the Cross-File-First setting of RepoBench-C-8k, maintaining the other settings as in the paper.
>
> Table R.3: Performance comparisons of DeepSeek-Coder 6.7B on contextually related insertion experiments
> | Language | Method                 | 5 lines       | 7 lines       | 11 lines      | 13 lines      | 15 lines      |   |   |   |
> |----------|------------------------|---------------|---------------|---------------|---------------|---------------|---|---|---|
> |          |                        | EM/ES         | EM/ES         | EM/ES         | EM/ES         | EM/ES         |   |   |   |
> | Python   | Full-recomputation     | 28.95 / 70.11 | 28.95 / 70.11 | 28.95 / 70.11 | 28.95 / 70.11 | 28.95 / 70.11 |   |   |   |
> | Python   | Conflict Fast Encoding | 7.92 / 41.56  | 6.39 / 38.56  | 4.78 / 34.91  | 4.11 / 33.34  | 3.43 / 32.15  |   |   |   |
> | Python   | PIE                    | 27.4 / 69.10  | 27.23 / 68.94 | 27.04 / 68.8  | 26.92 / 68.8  | 26.86 / 68.63 |   |   |   |
> |          |                        |               |               |               |               |               |   |   |   |
> | Java     | Full-recomputation     | 32.51 / 75.56 | 32.51 / 75.56 | 32.51 / 75.56 | 32.51 / 75.56 | 32.51 / 75.56 |   |   |   |
> | Java     | Conflict Fast Encoding | 7.92 / 41.56  | 6.39 / 38.56  | 4.78 / 34.91  | 4.11 / 33.34  | 0.74 / 5.57   |   |   |   |
> | Java     | PIE                    | 30.04 / 74.46 | 30.00 / 74.44 | 30.13 / 74.47 | 29.87 / 74.38 | 29.76 / 74.29 |   |   |   |
>
> As shown in the Table R.3, PIE still maintains its performance compared to full recomputation on longer edits and contextually related edits.
>
> **Regarding Weakness 3: Dependency on RoPE**
>
> PIE indeed depends on RoPE. However, given that RoPE is the most widely used positional encoding in LLMs (e.g., LLaMA[3,4,5], PaLM[6], DeepSeek[7], Mistral[8,9], Phi[10,11], Baichuan[12], Qwen[13], Grok[14], Yi[15]), our method is compatible with a broad range of models.
>
> [3] Touvron H, Lavril T, Izacard G, et al. Llama: Open and efficient foundation language models[J]. arXiv preprint arXiv:2302.13971, 2023.
>
> [4] Touvron H, Martin L, Stone K, et al. Llama 2: Open foundation and fine-tuned chat models[J]. arXiv preprint arXiv:2307.09288, 2023.
>
> [5] Dubey A, Jauhri A, Pandey A, et al. The llama 3 herd of models[J]. arXiv preprint arXiv:2407.21783, 2024.
>
> [6] Chowdhery A, Narang S, Devlin J, et al. Palm: Scaling language modeling with pathways[J]. Journal of Machine Learning Research, 2023, 24(240): 1-113.
>
> [7] Guo D, Zhu Q, Yang D, et al. DeepSeek-Coder: When the Large Language Model Meets Programming--The Rise of Code Intelligence[J]. arXiv preprint arXiv:2401.14196, 2024.
>
> [8] Jiang A Q, Sablayrolles A, Mensch A, et al. Mistral 7B[J]. arXiv preprint arXiv:2310.06825, 2023.
>
> [9] Jiang A Q, Sablayrolles A, Roux A, et al. Mixtral of experts[J]. arXiv preprint arXiv:2401.04088, 2024.
>
> [10] Gunasekar S, Zhang Y, Aneja J, et al. Textbooks are all you need[J]. arXiv preprint arXiv:2306.11644, 2023.
>
> [11] Li Y, Bubeck S, Eldan R, et al. Textbooks are all you need ii: phi-1.5 technical report[J]. arXiv preprint arXiv:2309.05463, 2023.
>
> [12] Yang A, Xiao B, Wang B, et al. Baichuan 2: Open large-scale language models[J]. arXiv preprint arXiv:2309.10305, 2023.
>
> [13] Bai J, Bai S, Chu Y, et al. Qwen technical report[J]. arXiv preprint arXiv:2309.16609, 2023.
>
> [14] https://github.com/xai-org/grok-1
>
> [15] Young A, Chen B, Li C, et al. Yi: Open foundation models by 01. ai[J]. arXiv preprint arXiv:2403.04652, 2024.

---

> > ### Author Response · Authors · 2024-11-19
> > **Response to Question 2, 4, 5 and 6**
> >
> > **Regarding Question 2: performance on non-code related tasks where the semantic relationship is important**
> >
> > Our work focuses on real-time code editing. For tasks where semantic relationships are crucial, we conduct experiments on HumanEval[1] and its C++ version from HumanEval-X[2] (see Table R.2 above). It's important to note that the context (prompt) in HumanEval contains only a few lines (an average of 15 lines), so the edits significantly affect semantics, validating the effectiveness of PIE. We also would like to study more non-code related tasks in the next version.
> >
> > **Regarding Question 4: Is there any memory overhead of the proposed approach?**
> > Suppose the total sequence length is $n$, an edit occurs at position $j$, the dimensionality of the key vectors is $d$, the number of attention heads is $h$, and $16-bit$ precision is used (so each element occupies $2$ bytes).
> >
> > The memory overhead for PIE is given by:
> >
> > Memory overhead=$(n−j)×d×h×2$
> >
> > Taking DeepSeek-Coder 6.7B as an example, where $d = 128$ and $h = 32$, the memory overhead for PIE becomes:
> >
> > $(n−j)×128×32×2bytes=(n−j)×8 KB$
> >
> > This overhead is sufficiently small to be considered negligible in practical applications.
> >
> > **Regarding Question 5: Does this approach lead to any cumulative errors if you continue to update KV cache based on PIE?**
> >
> > In Table R.3 above, we report the performance of PIE on longer edits. As the number of edited tokens increases, PIE successfully retains most of its performance, further validating the effectiveness of PIE.
> >
> > **Regarding Question 6: The average scores of 3 experiments are reported, could you also report the standard deviation?**
> > The results across different runs are consistent, showing minimal variation. Therefore, we chose to omit the standard deviation in our reporting to maintain clarity. For example, the exact match of PIE of DeepSeek-Coder 6.7B on XF-F python insertion experiments across three different runs is 28.80, 28.83, and 28.87, resulting in a negligible standard deviation of 0.03.
> >
> > Thank you for your comprehensive review. We have carefully replied to your questions and will improve the paper based on your feedback. Please let us know if you need any additional clarification.

---

> ### Author Response · Authors · 2024-11-23
>
> Dear reviewer,
>
> Thank you for your constructive review and comments! We have provided point-to-point responses to your review. Considering the author-reviewer discussion is ending very soon, could you kindly take a look at our responses and let us know your feedback? We are happy to discuss more if you still have any concerns. Thank you again and look forward to your feedback!

---

> > ### Comment · Reviewer_Yphe · 2024-11-23
> >
> > Thank you for providing detailed answers and additional results. Most of my concerns have been addressed. I have raised my score.

---

> > > ### Author Response · Authors · 2024-11-23
> > >
> > > Dear reviewer,
> > >
> > > We greatly appreciate your positive feedback. We noticed there might be a small technical issue, as the updated score you mentioned is not yet reflected in the system. Please feel free to verify this at your convenience.

---

> > > > ### Comment · Reviewer_Yphe · 2024-11-23
> > > >
> > > > I can see the updated score. Thanks!

---

### Official Review · Reviewer_qWyU · 2024-11-03

**Soundness:** 1
**Presentation:** 3
**Contribution:** 2
**Rating:** 3
**Confidence:** 4

**Summary:**

This paper introduces a technique to update the rotary positional encoding when a small part of the context tokens are updated. It aims to optimize computational efficiency in real-time editing scenarios. The authors show that by fixing the positional encoding alone, they were able to retain a performance that is almost as good as fully re-encoding the context on a left-to-right code generation task, but with considerably less computational cost.

**Strengths:**

The paper effectively outlines the real-time editing problem and clearly describes the mathematical foundation for PIE based on rotary positional encoding.

**Weaknesses:**

**Limited Technical Novelty**: The mathematical derivation is relatively straightforward, stemming directly from rotary positional encoding's relative nature, without additional innovation or complexity.

**Unrealistic Setting for Interactive Editing**:
 * Random Edits Only: The experimental setup evaluates PIE on random edits, which does not align with realistic real-time editing workflows, where temporally or contextually related edits are more common (e.g., editing a function signature and then updating its call sites). PIE’s simplifications may therefore be less applicable to typical usage patterns.
 * Single-Edit Evaluation: The paper evaluates PIE in a single-edit scenario, overlooking the potential for accumulated errors in multi-edit settings. In practical applications, users often make multiple edits, which could introduce drift in the positional encoding without full recomputation.
 * Left-to-Right Only: Evaluations are limited to left-to-right generation, omitting fill-in-the-middle (FIM) generation, a task relevant in code editing where users may modify code segments in the middle of sequences. Without this, it is unclear how PIE would perform in varied editing tasks.

**Unconvincing Conclusions Due to Limited Evaluation Scope**: Given the unrealistic evaluation settings, the claim that PIE can retain performance by adjusting positional encoding alone is unconvincing. By testing only on random edits, the experiments fail to address cases where contextual dependencies (e.g., edits that affect other tokens' relevance) might demand full recomputation. This risks overstating PIE’s applicability to real-world editing scenarios. To build a more compelling case, I recommend:
 * Evaluating PIE on real user edit sequences rather than synthetic random edits.
 * Restricting comparisons with full recomputation to cases where edited tokens impact final target tokens meaningfully (e.g., by verifying that removing or masking these edited tokens affects the target token prediction likelihood).
 * Including a special baseline where the pre-edit context is reused without modification, establishing a zero-cost apporach for comparison.

**Lack of Multi-Edit and Accumulated Error Analysis**: With each edit, additional errors will enter the encoding under the proposed technique, but the paper provides no analysis of error accumulation across multiple edits. Without such discussion, it’s unclear when a full recomputation might be needed to reset the encoding.

**Lack of Fill-in-the-Middle Evaluation**: Evaluations are limited to left-to-right generation, omitting fill-in-the-middle (FIM) generation, which is more relevant to the interactive coding assistant scenarios mentioned by the paper.

**Questions:**

Could you provide performance results for a baseline that reuses the pre-edit context without modification for making suggestions? This zero-cost approach would be helpful to compare against Full-recomputation in your benchmark.

---

> ### Author Response · Authors · 2024-11-19
>
> Thanks for your careful review and suggestions. We respond to each of your concerns as below.
>
> **Regarding Unrealistic Setting for Interactive Editing and Unconvincing Conclusions Due to Limited Evaluation Scope**
>
> We opted for a random edit setting due to the lack of real-world datasets that capture real-time user editing with corresponding labels. Since no such datasets exist, we leveraged the current dataset and constructed a random edit scenario. This allows us to ensure comprehensive coverage of diverse editing scenarios without making assumptions or imposing restrictions on user input. In real-world applications, user inputs are highly varied and unpredictable, and random edits, which can occur at any point in the code, provide a more general and robust evaluation of our method across a wide range of potential situations.
>
> Additionally, following your valuable suggestions, we incorporated a reuse baseline and conducted experiments in more diverse editing conditions, specifically on: (1) multiple edits and contextually related edits, and (2) fill-in-the-middle (FIM) generation, as detailed below.
>
> 1. **Multiple edits and contextually related edits**
>
> To simulate multi-edit scenarios, we increase the number of edited lines. We also design a contextually related insertion setting, where we use Edit Distance to identify lines in the context that are similar to the target line, simulating user insertions. We conduct experiments (see Table R.1) on the Cross-File-First setting of RepoBench-C-8k, keeping all other settings the same as in the paper.
>
> Table R.1: Performance of DeepSeek-Coder 6.7B on contextually-related insertion experiments
> | Language | Method                 | 5 lines       | 7 lines       | 11 lines      | 13 lines      | 15 lines      |   |   |   |
> |----------|------------------------|---------------|---------------|---------------|---------------|---------------|---|---|---|
> |          |                        | EM/ES         | EM/ES         | EM/ES         | EM/ES         | EM/ES         |   |   |   |
> | Python   | Full-recomputation     | 28.95 / 70.11 | 28.95 / 70.11 | 28.95 / 70.11 | 28.95 / 70.11 | 28.95 / 70.11 |   |   |   |
> | Python   | Conflict Fast Encoding | 7.92 / 41.56  | 6.39 / 38.56  | 4.78 / 34.91  | 4.11 / 33.34  | 3.43 / 32.15  |   |   |   |
> | Python   | Reuse baseline         | 25.58 / 68.23 | 25.29 / 68.03 | 25.23 / 67.95 | 25.13 / 67.83 | 24.71 / 67.74 |   |   |   |
> | Python   | PIE                    | **27.4** / **69.10**  | **27.23** / **68.94** | **27.04** / **68.8**  | **26.92** / **68.8**  | **26.86** / **68.63** |   |   |   |
> |          |                        |               |               |               |               |               |   |   |   |
> | Java     | Full-recomputation     | 32.51 / 75.56 | 32.51 / 75.56 | 32.51 / 75.56 | 32.51 / 75.56 | 32.51 / 75.56 |   |   |   |
> | Java     | Conflict Fast Encoding | 7.92 / 41.56  | 6.39 / 38.56  | 4.78 / 34.91  | 4.11 / 33.34  | 0.74 / 5.57   |   |   |   |
> | Java     | Reuse basline          | 25.99 / 73.29 | 25.72 / 73.17 | 25.18 / 73.03 | 25.15 / 72.99 | 24.99 / 72.84 |   |   |   |
> | Java     | PIE                    | **30.04** / **74.46** | **30.00** / **74.44** | **30.13** / **74.47** | **29.87** / **74.38** | **29.76** / **74.29** |   |   |   |
>
> The reuse baseline has been newly incorporated following your suggestion. The results in Table R.1 demonstrate that PIE is robust in handling longer edits and outperforms the baseline methods, approaching the performance of full recomputation.
>
> To further address your concern about the effectiveness of PIE,  we conduct experiments on code generation tasks (as suggested by Reviewer Bo49), using HumanEval [1] and its C++ version from HumanEval-X [2]. We randomly choose three consecutive lines in the prompt as the user inserted code and report Pass@1 for different methods. It's important to note that the context (prompt) in HumanEval consists of only a few lines (with an average of 15 lines), making each edit semantically significant. The results are shown in Table R.2 below:
>
> Table R.2: Performance of DeepSeek-Coder 6.7B on HumanEval and HumanEval-X
> | Model               | Method                 | Python (Pass@1) | C++ (Pass@1) |
> |---------------------|------------------------|-----------------|--------------|
> | DeepSeek-Coder 6.7B | Full-recomputation     | 49.4            | 50.3         |
> | DeepSeek-Coder 6.7B | Conflict Fast Encoding | 11.0            | 15.5         |
> | DeepSeek-Coder 6.7B | Reuse Baseline         | 17.7            | 29.2         |
> | DeepSeek-Coder 6.7B | PIE                    | **43.9**            | **49.7**         |
> |                     |                        |                 |              |

---

> ### Author Response · Authors · 2024-11-19
>
> The results in Table R.2 demonstrate that PIE significantly outperforms the baseline methods, approaching the performance of full recomputation. Given the importance of these edits in such short contexts, updating the KV cache with PIE becomes essential for maintaining high prediction accuracy, whereas the performance of the reuse baseline drops substantially.
>
> 2. **Fill-in-the-middle (FIM) generation**
>
> Since RepoBench doesn't provide the right context, we use the CrossCodeEval dataset [3] for left-to-right evaluation. We utilize the complete left context and prompt, truncating the right context to a maximum of 40 lines to construct the fill-in-the-middle task. We use Edit Distance to identify five consecutive lines in the left context that are similar to the target line, simulating user insertions. The results are provided in Table R.3 below.
>
> Table R.3:  Performance of DeepSeek-Coder 6.7B on Fill-in-the-middle task
> |Language        | Method                 | 5 lines       |
> |--------|------------------------|---------------|
> |        |                        | EM / ES       |
> | Python | Full-recomputation     | 11.97 / 55.71 |
> | Python | Conflict Fast Encoding | 0.6 / 15.32   |
> | Python | Reuse                  | 8.44 / 50.16  |
> | Python | PIE                    | **10.24** / **53.24** |
> |        |                        |               |
>
>
> Table R.3 shows that PIE still maintains performance in the Fill-in-the-middle task, which is consistent with the left-to-right task.
>
> **Regarding Limited Technical Novelty**
>
> Our paper's primary contribution is PIE, an simple yet effective approach designed for real-time code editing. While previous research has largely overlooked the simultaneous modification of context and model representations (KV cache) in dynamic code editing scenarios, our work directly addresses this gap. By resolving temporal confusion in real-time editing, PIE offers a novel solution for improved efficiency in dynamic environments.
> We believe that by resolving temporal confusion in real-time code editing,  PIE opens new pathways for improving LLM efficiency in dynamic environments, contributing a novel and impactful solution to the field.
>
>
> We appreciate your thorough review of our paper. We have carefully replied to your questions and will improve our paper according to your suggestions. We look forward to your re-evaluation of our submission based on our responses, and we are also happy to go into more detail regarding any further questions.
>
> [1] Chen M, Tworek J, Jun H, et al. Evaluating large language models trained on code[J]. arXiv preprint arXiv:2107.03374, 2021.
>
> [2] Zheng Q, Xia X, Zou X, et al. Codegeex: A pre-trained model for code generation with multilingual benchmarking on humaneval-x[C]//Proceedings of the 29th ACM SIGKDD Conference on Knowledge Discovery and Data Mining. 2023: 5673-5684.
>
> [3] Ding Y, Wang Z, Ahmad W, et al. Crosscodeeval: A diverse and multilingual benchmark for cross-file code completion[J]. Advances in Neural Information Processing Systems, 2024, 36.

---

> ### Author Response · Authors · 2024-11-23
>
> Dear reviewer,
>
> Thank you for your constructive review and comments! We have provided point-to-point responses to your review. Considering the author-reviewer discussion is ending very soon, could you kindly take a look at our responses and let us know your feedback? We are happy to discuss more if you still have any concerns. Thank you again and look forward to your feedback!

---

> ### Comment · Reviewer_qWyU · 2024-11-24
>
> > In real-world applications, user inputs are highly varied and unpredictable, and random edits, which can occur at any point in the code, provide a more general and robust evaluation of our method across a wide range of potential situations.
>
> I respectfully disagree with this claim. Realistic edits are often temporally and contextually correlated, as users tend to modify interdependent parts of the code. For example, after editing a function signature, a user is likely to update related call sites. In such cases, the ability to attend to newly edited code is essential for maintaining prediction accuracy. Random edits, by contrast, fail to capture these dependencies, making them a poor representation of real-world editing scenarios and reducing the relevance of your evaluation.
>
> > Additionally, following your valuable suggestions, we incorporated a reuse baseline and conducted experiments in more diverse editing conditions, specifically on: (1) multiple edits and contextually related edits, and (2) fill-in-the-middle (FIM) generation, as detailed below.
>
> Thank you for including the reuse baseline and additional experiments. However, the performance of this new reuse baseline appears to significantly outperform the original baselines used in your paper. This undermines the original baseline's relevance and raises concerns about its suitability as a comparison point. I strongly recommend incorporating the reuse baseline across all new results provided in your replies to other reviewers. The fact that the reuse baseline—essentially a "do nothing" approach—captures most of the gains PIE demonstrated over Conflict Fast Encoding further questions the significance of PIE’s remaining improvements. This raises the possibility that these improvements may stem from benchmarks crafted in ways that favor PIE, rather than inherent advantages of the method itself.
>
> I also don’t feel that my concerns regarding the single-edit limitation have been adequately addressed by the new results that simply increase the number of edited lines. Consider the definition-call site example again: a user might first edit a function definition and then proceed to edit all associated call sites. Under PIE's attention pattern, the transformed context would be able to attend neither the newly edited definition nor any of the newly edited call sites, making it very difficult (if not impossible) to correctly predict the remaining call sites that needed an update. If your evaluation instead increases the size of a single edit site or simply picks more random edit sites, this artificially increases the chance that at least one new call site remains untouched and leaks the correct calling pattern into the context. This, however, reflects an unrealistic evaluation scenario and does not align with how PIE would behave in practical usage. In summary, PIE’s inability to allow context tokens to attend to newly edited tokens seems like a critical limitation. making it highly unlikely for PIE to match the performance of full recomputation in realistic usages. I encourage the authors to address this gap with more concrete examples or additional analyses.
>
> That said, I acknowledge that the inclusion of the reuse baseline and the new experiments have strengthened the overall results. Assuming that the reuse baseline will be incorporated into all reported results, I am raising my score from 1 to 3 to reflect these improvements. However, I remain concerned about the broader applicability of PIE in realistic editing scenarios.

---

> ### Author Response · Authors · 2024-11-27
> **Response to the reuse baseline and multi-place edits**
>
> Dear Reviewer qWyU,
>
> Thank you for your further comments and for raising specific concerns about the broader applicability of PIE in realistic editing scenarios. We appreciate the opportunity to address these points in detail.
>
> **1. Regarding the reuse baseline compared to PIE**
>
> We appreciate your suggestion to incorporate the reuse baseline as a comparison point. While the reuse baseline may capture some performance gains due to the redundancy in code, it performs poorly when user edits involve **highly important and semantically significant changes**. This limitation is particularly evident in HumanEval (Table R.2), where the context (prompt) consists of only a few lines (with an average of 15 lines), making each edit semantically significant.
>
> To further expand on this, we conduct additional experiments using a more challenging setting, where nine consecutive lines were chosen as the user-inserted code. The results for this setup are shown in Table R.4 below:
>
> Table R.4: Performance of DeepSeek-Coder 6.7B on HumanEval and HumanEval-X (9 lines)
> | Model               | Method                 | Python (Pass@1) | C++ (Pass@1) |
> |---------------------|------------------------|-----------------|--------------|
> | DeepSeek-Coder 6.7B | Full-recomputation     | 49.4            | 50.3         |
> | DeepSeek-Coder 6.7B | Conflict Fast Encoding | 25              | 26.7         |
> | DeepSeek-Coder 6.7B | Reuse Baseline         | 4.3             | 12.4         |
> | DeepSeek-Coder 6.7B | PIE                    | **45.7**            | **48.4**         |
> |                     |                        |                 |              |
>
> The results in Table R.4 clearly demonstrate that the **reuse baseline essentially fails in this task, dramatically underperforming** compared to Conflict Fast Encoding and PIE.
>
> **2. Regarding the single-edit setting and multi-place edits**
>
> You raise the valuable example of a user editing a function definition followed by updates to all associated call sites. To simulate this scenario, we treat each line in the context as a potential site and use Edit Distance to identify multiple sites in the context that are similar to the target line, thereby simulating multi-place insertions. We conduct experiments in this contextually related multi-place editing setting on the Cross-File-First task of RepoBench-C-8k. The results are shown in Table R.5 below:
>
> Table R.5: Performance of DeepSeek-Coder 6.7B on contextually-related multi-place insertion experiments
> | Language | Method                 | 5 sites       | 7 sites        | 11 sites      | 13 sites      | 15 sites      |
> |----------|------------------------|---------------|----------------|---------------|---------------|---------------|
> |          |                        | EM / ES       | EM / ES        | EM / ES       | EM / ES       | EM / ES       |
> | Python   | Full-recomputation     | 28.95 / 70.11 | 28.95 / 70.11  | 28.95 / 70.11 | 28.95 / 70.11 | 28.95 / 70.11 |
> | Python   | Conflict Fast Encoding | 7.43 / 39.95  | 5.83 / 37.0    | 4.06 / 33.02  | 3.48 / 31.71  | 2.97 / 30.59  |
> | Python   | Reuse baseline         | 24.61 / 67.43 | 24.44 / 67.17  | 23.84 / 66.77 | 23.62 / 66.63 | 23.49 / 66.55 |
> | Python   | PIE                    | 26.97 / 68.85 | 26.77 / 68.67  | 26.33 / 68.34 | 26.15 / 68.22 | 25.9 / 68.23  |
> |          |                        |               |                |               |               |               |
> | Java     | Full-recomputation     | 32.51 / 75.56 | 32.51 / 75.56  | 32.51 / 75.56 | 32.51 / 75.56 | 32.51 / 75.56 |
> | Java     | Conflict Fast Encoding | 1.69 / 10.91  | 1.06 / 7.66    | 0.43 / 4.26   | 0.28 / 3.36   | 0.22 / 2.81   |
> | Java     | Reuse basline          | 24.46 / 72.35 | 23.99 / 72.05  | 23.11 / 71.69 | 22.97 / 71.56 | 22.75 / 71.38 |
> | Java     | PIE                    | 29.56 / 74.14 | 29.26 / 73.91  | 29.03 / 73.81 | 28.83 / 73.68 | 28.56 / 73.59 |
> |          |                        |               |                |               |               |               |
>
> The results in Table R.5 demonstrate that PIE is robust in handling multi-place edits and outperforms the baseline methods.
>
> We hope these additional experiments and analyses address your concerns and provide further evidence of PIE’s robustness and applicability to real-world editing scenarios. Thank you again for your thoughtful review and suggestions. We are happy to provide further details or conduct additional experiments if you have any further questions.

---

> ### Comment · Reviewer_qWyU · 2024-12-02
>
> Thank you for your detailed response and for conducting additional experiments to address my concerns. I appreciate the effort in expanding the evaluation to include the reuse baseline and exploring multi-place edits. However, I have reservations about the realism of the new settings you proposed.
>
> If I understood correctly, the multi-place editing setup appears to simulate edits by removing insertion points from the final version of the code and treating the resulting broken code as the "original code" for the reuse baseline. This does not reflect how code is typically edited iteratively over time or incrementally updated across related call sites. By using broken code as input for the reuse baseline, the comparison feels artificially skewed in favor of PIE and does not fairly evaluate the reuse baseline's performance in a realistic iterative editing workflow.
>
> Real-world editing often involves scenarios where users progressively refine their code, and the ability to attend to new edits (e.g., function definitions and their associated call sites) is critical. This evaluation setup does not adequately capture those dynamics, which remain a key concern about PIE's applicability.
>
> Given these issues, I am unable to update my current score. I still appreciate the additional experiments and analyses, but I encourage further exploration of settings that more accurately reflect real-world iterative editing processes. Thank you again for engaging with my feedback and providing thoughtful responses.

---

> ### Author Response · Authors · 2024-12-02
> **Response to multi-place editing setup and progressive refinement**
>
> Dear Reviewer qWyU,
>
> Thank you for your continued engagement and for raising concerns about the realism of our multi-place editing setup. We appreciate the opportunity to clarify our methodology and address your points in detail.
>
> **1. Clarification of the Multi-Place Editing Setup**
>
> You correctly understood our multi-place editing setup: it simulates edits by removing insertion points from the final version of the code and treating the resulting broken code as the "original code".
>
> However, we respectfully disagree with your statement that this comparison is "artificially skewed in favor of PIE" and does not fairly evaluate the reuse baseline’s performance.
>
> Regardless of whether edits are applied progressively or simultaneously, the reuse baseline inherently fails to capture the changes introduced by the edits, as it relies entirely on the original, unmodified context. This limitation is fundamental to the reuse baseline and is not influenced by how edits are simulated or applied. By definition, it cannot adapt to the edited context and will yield the same output regardless of the editing workflow. Therefore, we believe that our comparison remains fair and highlights the limitations of the reuse baseline relative to PIE.
>
> Moreover, we would like to highlight the results in Table R.2 and Table R.4, where the reuse baseline essentially fails, dramatically underperforming compared to both Conflict Fast Encoding and PIE. Unlike the reuse baseline and conflict fast encoding, PIE effectively utilizes the inserted code, enabling it to retain most of the performance and achieve results much closer to full recomputation.
>
> **2. Progressive Refinement**
>
> You also mentioned that real-world editing often involves progressive refinement, where users iteratively update their code. To address this, we provide the following clarification:
>
> In the insertion experiments, if the edits are applied progressively (i.e., each insertion is made, followed by PIE updates) or simultaneously (i.e., all edits are made at once, and PIE is applied afterward), the results will be identical as long as the insertions are orderly (e.g., from the beginning of the context to the end).
>
> If the insertions are unordered or dispersed, we agree that there may be slight differences between progressive and simultaneous updates. However, we argue that this difference is likely to be small, as the core mechanism of PIE—resolving temporal confusion in the KV cache—remains consistent.
>
> Thank you for your thoughtful feedback. We hope this response clarifies our experimental design and addresses your concerns about PIE's applicability in realistic scenarios. We remain committed to further improving our evaluation methodology and welcome any additional suggestions you may have.
>
> Best regards,
>
> Authors

---

### Meta-Review · Area_Chair_NCx4 · 2024-12-22

**Metareview:**

This paper investigates an interesting question and presents Positional Integrity Encoding (PIE) as an inexpensive alternative to re-encoding the entire KV cache during LLM decoding, specifically for code-related tasks. The reviewers agree that the results are impressive and demonstrate significant contributions to the field. However, several suggestions for improvement remain, including issues with presentation, clarity in technical discussions, and a need for more comparisons with other contenders as well as deeper experimental analysis. The authors are encouraged to revise the paper to enhance its impact carefully.

**Additional Comments On Reviewer Discussion:**

The reviewers agree that the results are impressive and demonstrate significant contributions to the field. However, several suggestions for improvement remain, including issues with presentation, clarity in technical discussions, and a need for more comparisons with other contenders as well as deeper experimental analysis.

---

### Decision · Program_Chairs · 2025-01-22

Accept (Poster)